

# HIDRA-D: deep-learning model for dense sea level forecasting using sparse altimetry and tide gauge data

Marko Rus[2], Matjaž Ličer[1,3,★], and Matej Kristan[2,★]

[1]Office for Meteorology, Hydrology and Oceanography, Slovenian Environment Agency, Ljubljana, Slovenia
[2]Visual Cognitive Systems Lab, Faculty of Computer and Information Science, University of Ljubljana, Ljubljana, Slovenia
[3]Marine Biology Station, National Institute of Biology, Piran, Slovenia
[★]These authors contributed equally to this work.

**Correspondence:** Matjaž Ličer (matjaz.licer@gov.si)

**Abstract.**

This paper introduces HIDRA-D, a novel deep-learning model for basin scale dense (gridded) sea level prediction from in situ tide gauge data. Accurate sea level prediction is crucial for coastal risk management, marine operations, and sustainable development. While traditional numerical ocean models are computationally expensive, especially for probabilistic forecasts over many ensemble members, HIDRA-D offers a faster, numerically cheaper, observation-driven alternative. Unlike previous HIDRA models (HIDRA1, HIDRA2 and HIDRA3) that focused on point predictions at tide gauges, HIDRA-D provides dense, two-dimensional, gridded sea level forecasts. The core innovation lies in a new algorithm that effectively leverages sparse and unevenly distributed satellite altimetry data in combination with tide gauge observations, to learn the complex basin-scale dynamics of sea level. HIDRA-D achieves this by integrating a HIDRA3 module for point predictions at tide gauges with a novel Dense decoder module, which generates low-frequency spatial components of the sea level field in the Fourier domain, whose Fourier inverse is an hourly sea level forecast over a 3-day horizon. Evaluation in the Adriatic demonstrates that HIDRA-D significantly outperforms the NEMO general circulation model, achieving a 28.0 % reduction in mean absolute error when compared to satellite sea-level anomaly (SLA) data. However, while HIDRA-D performs well in open waters, leave-one-out cross-validation at tide gauges indicates limitations in areas with complex bathymetry, such as the Neretva estuary located in a narrow bay, and in regions with sparse SLA data, like the northern Adriatic. Importantly, the model shows robustness to spatially-limited tide gauge coverage, maintaining acceptable performance even when trained using data from distant stations. This suggests its potential for broader applicability in areas with limited in situ observations.

## 1 Introduction

The ability to accurately predict sea levels has become increasingly vital for proactive planning and mitigation efforts across diverse sectors. Reliable forecasts are crucial for managing coastal risks, optimizing marine operations, and supporting sustainable development in the face of a changing climate. However, predicting sea levels is an inherently complex task. The ocean is a dynamic system influenced by a multitude of interacting factors, including atmospheric pressure, winds, tides, and currents, all operating across a wide range of spatial and temporal scales. Traditional numerical models (Umgiesser et al.,





2022; Ferrarin et al., 2020; Madec, 2016) provide an invaluable spatiotemporal insight into the state of the ocean, ranging

from density fields to sea levels. This insight however comes at a high numerical price and even then the models often crucially depend on numerically-intensive data assimilation (Bajo et al., 2023) and local bias corrections to remain close to observations. This is especially true in coastal regions characterized by complex bathymetry and rapidly varying hydrodynamic conditions (Umgiesser et al., 2022).

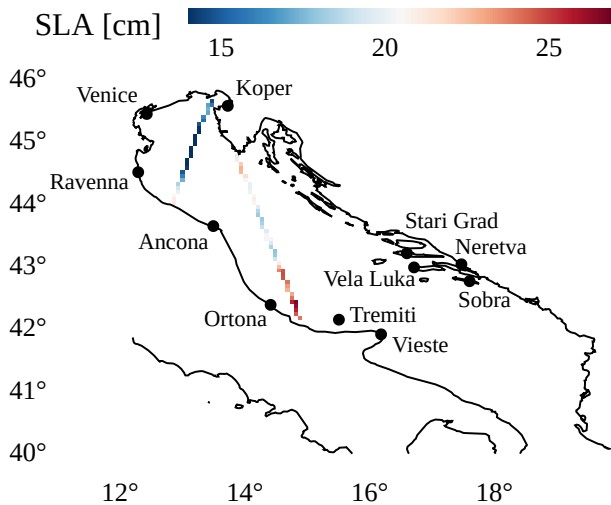

**Figure 1.** The region of our study, the Adriatic Sea, with 11 tide gauge locations depicted by black dots and with the two satellite sea-level anomaly (SLA) swaths from June 6, 2019, shown.

The inherent uncertainties of the model initial conditions, boundary conditions, and even the representation of physical

processes themselves necessitate a probabilistic approach when modeling using a numerical model (Ferrarin et al., 2023; Bernier and Thompson, 2015; Mel and Lionello, 2014). Instead of a single, deterministic prediction, numerical models rely on ensemble modeling, generating multiple simulations with slightly varying parameters to capture the envelope of possible sea level outcomes. While this approach improves robustness, it comes at a steep price. The computational cost increases with the number of ensemble members, making high-resolution, real-time forecasting computationally expensive. To overcome this

computational barrier, our research has been focused on developing a family of fast deep learning algorithms for sea level predictions, collectively known as HIDRA (Rus et al., 2025d, 2023; Žust et al., 2021), designed to drastically reduce the computational cost of sea level forecasting while maintaining or even exceeding the accuracy of traditional numerical models.

The evolution of HIDRA has been driven by a continuous process of refinement and expansion. The initial HIDRA1 model (Žust et al., 2021) demonstrated that a deep learning model could predict sea surface height (SSH) at a single tide gauge

location (Koper, Slovenia) with improved accuracy and a million times faster than our operational numerical model NEMO GCM (Ličer et al., 2020). This indicated the fundamental feasibility of the deep learning approach to sea level modeling. The performance was subsequently improved by the HIDRA2 (Rus et al., 2023) model by a large margin. However, HIDRA2



depends on a large observational dataset at the prediction location, making it unreliable for areas with sparse data and it cannot deliver sea level predictions during tide gauge sensor failure. HIDRA3 (Rus et al., 2025d) addressed the data sparsity limitation

by enabling multi-point predictions across a network of tide gauges. Furthermore, HIDRA3 can infer future sea levels also at locations with sensor failures by leveraging data from neighboring operational stations. This significantly improves prediction robustness and resilience. However, predictions of HIDRA3 are nevertheless confined to the locations of existing tide gauges and cannot predict sea levels offshore at an arbitrary point in the basin.

Building upon this foundation, we introduce HIDRA-D, which represents a fundamental shift from point-based predictions

at sparse locations to dense, gridded, two-dimensional sea level forecasts across the entire ocean basin. Unlike previous HIDRA versions, which were trained solely using tide gauge data, HIDRA-D incorporates satellite altimetry observations (sea level anomaly, or SLA) to learn the spatial patterns of sea level. This presents a unique challenge, as SLA data, while valuable, is extremely sparse and unevenly distributed, both spatially and temporally (see Fig. 1). While advancements exist in interpolating irregularly sampled satellite data, often focusing on variables with greater coverage like sea surface temperature (Barth et al.,

2020, 2022) or employing wide-swath altimetry (Fablet et al., 2021; Beauchamp et al., 2023), we specifically leverage along-track sea level altimetry for its extended historical availability.

In the text that follows, SLA (sea level anomaly) will be used to denote remote satellite altimetry observations. It is important to note that in this study, when we refer to SLA, we are actually referring to absolute dynamical topography (ADT), which includes a mean dynamic topography (MDT) component. SSH (sea surface height) will be used to denote local tide gauge

observations.

As a rule, SLA measurements from the satellite altimeter are not calibrated with the SSH measurements at different tide gauges and, furthermore, the tide gauges are often not calibrated between each other, each reporting the sea level values relative to their local vertical datum. In this paper we propose a novel formulation that casts tide gauge and SLA intercalibration as part of the learning problem. At the time of application, HIDRA-D considers only the sparse tide gauge inputs and atmospheric

data, while providing long-horizon dense SLA predictions. As a side-product, HIDRA-D provides for all tide gauges an inter-calibration transformation to the satellite-referenced SLA, thus mutually-calibrating them.

The remainder of this paper is structured as follows. Section 2 provides a detailed description of the HIDRA-D architecture, including the integration of the HIDRA3 module, the novel Dense encoder module and a new method for calibrating tide gauge measurements with the satellite SLA measurements. We also detail the training, validation, and testing datasets, along

with implementation specifics. Section 3 presents a comprehensive evaluation of HIDRA-D. We compare its dense sea level forecasts against both SLA satellite observations and the well-established Copernicus NEMO general circulation model for the Mediterranean basin (Clementi et al., 2021). We also assess its performance at arbitrary coastline locations and analyze robustness to data limitations.



## 2 HIDRA-D: deep learning basin-scale sea level forecasting model

### 2.1 Training and testing datasets

Our objective is predicting the temporal evolution of sea level in the Adriatic basin on a two-dimensional grid (i.e., dense prediction) with a 3-day horizon at hourly resolution. The horizontal and vertical grid sizes are $H = 94$ and $W = 115$ respectively, spanning a geographical region bounded by latitudes 40.00°N to 45.87°N and longitudes 12.20°E to 18.85°E. The input to the prediction model consists of past 72 h of available SSH observations from $N = 11$ Adriatic tide gauges together with their
respective astronomic tides, as well as the past and future 72 h of gridded geophysical variables obtained by the atmospheric and ocean numerical models.

The training time window spans two intervals: from 2000 to May 2019 and from 2021 to 2022. The testing time window covers the period from June 2019 to the end of 2020. This specific testing interval was selected due to the occurrence of numerous high sea level and coastal flood events in the northern Adriatic and matches the period chosen in our previous work
(Rus et al., 2025d).

The following tide gauges along the Adriatic coast are considered in this study for SSH measurements: Koper, Venice, Ancona, Ortona, Vieste, Neretva, Ravenna, Sobra, Stari Grad, Tremiti and Vela Luka (see Fig. 1). Their SSH availability ranges from 15 % to 90 % during years 2000–2022 (Rus et al., 2025d), which has to be accounted for during training and testing, as the model is required to cast predictions also when data from some tide gauges is missing. The SSH data was filtered
using the methods from Rus et al. (2025d) to eliminate three types of sensor errors: (i) sensor freeze, which results in a constant output value for an extended period of time, (ii) extreme outliers, and (iii) extreme jumps in the signals. For each location, the astronomical tides in 1-year intervals were computed using the UTIDE Tidal Analysis package for Python (Codiga, 2011).

Following Rus et al. (2025d), the sea level readings are classified as *low* if they fall below the 1st percentile and as *high* if they exceed the 99th percentile of the observed values at the given location (see Rus et al. (2025d) for exact thresholds). During
evaluation (Sect. 3.4), several metrics are computed separately for all sea level values and for the aforementioned extreme classes. This enables assessing the model's ability to predict both tails of the sea level distribution: (i) high values, which are crucial for coastal flood warnings, as well as (ii) low values, which may restrict marine traffic in the shallow northern region of the Adriatic basin.

For geophysical variables, we employed ERA5 reanalysis data (Hersbach et al., 2023) for training and ECMWF Ensemble
Prediction System (EPS) data (Leutbecher and Palmer, 2007) for evaluation. The reason for using the ERA5 data in training is that our prior research indicated improved performance in the multi-point (i.e., spatially sparse) prediction setup. However, the evaluation is carried out on the ECMWF EPS dataset to faithfully reflect the operational forecasting setup in which reanalysis is unavailable and ensemble forecasts are typically used. The following parameters from ERA5 reanalysis were used: (i) 10-meter winds, (ii) mean sea level air pressure, (iii) sea surface temperature (SST), (iv) mean wave direction, (v) mean wave period and
significant height of combined wind waves and (vi) the swell data. Following our previous work (Rus et al., 2025d), all input fields were spatially cropped to the Adriatic basin and subsampled to a $9 \times 12$ spatial grid.



For operational forecasting and evaluation, an ensemble of $n_{\text{ens}} = 50$ atmospheric members is used. HIDRA-D processes each member separately, generating 50 dense sea level forecasts, which are then averaged to produce the final prediction.

### 2.1.1 Sea level anomaly data

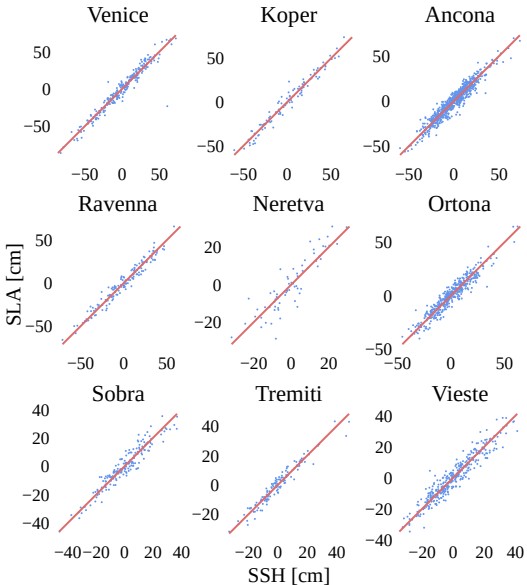

**Figure 2.** Correlations between satellite (SLA) and tide gauge (SSH) measurements at 9 locations. An SLA measurement is assigned to a tide gauge if it falls within 20 km of its location. The identity line is shown in red. Some degree of discrepancy between SLA and SSH measurements is observed, with an average absolute difference of 4.0 cm.

To enable supervised training of the network on the entire basin, the along-track sea level anomalies (SLA) from altimeter satellites were acquired from the Copernicus marine service product `SEALEVEL_EUR_PHY_L3_MY_008_061`. The SLA values are calculated relative to a 20-y mean (1993–2012) with a 1 Hz (∼7 km) sampling resolution. This dataset incorporates data from all available altimeter missions, including Sentinel-6A, Jason-3, Sentinel-3A, Sentinel-3B, Saral/AltiKa, Cryosat-2, Jason-1, Jason-2, Topex/Poseidon, ERS-1, ERS-2, Envisat, Geosat Follow-On, HY-2A, HY-2B, and others. The SLA values in this study are computed as the sum of variables SLA_filtered, ocean_tide, mean dynamic topography (MDT) and dynamic atmospheric correction (DAC). This is consistent with how SLA is treated in the CMEMS NEMO model for the purposes of data assimilation (Ali Aydogdu, CMCC, personal communication) and thus enables comparisons to the NEMO model (Clementi et al., 2021).

For comparisons with hourly measurements from our model or NEMO, the time of an SLA measurement was not rounded to the nearest hour, recognizing that sea level can change rapidly. Instead, hourly time series from our model or NEMO were linearly interpolated to the exact time of each SLA observation. The spatial locations of the SLA measurements were binned into a grid with size $94 \times 115$ equal to the spatial output of our model. In cases where multiple SLA measurements fall within



the same grid cell, the average of those measurements is taken. For visualization purposes, SLA values are adjusted to have the mean equal to zero.

To validate the satellite altimetry data and assess its accuracy in coastal regions, it is crucial to compare it with independent, in situ measurements. Tide gauges provide reliable SSH observations at specific coastal locations. Therefore, to verify the correlation between SLA and tide gauge SSH measurements, in Fig. 2 we present pairs of SLA and SSH measurements. An SLA measurement is assigned to a tide gauge if it falls within 20 km of its location. If multiple SLA measurements from the same satellite swath satisfy this condition, we retain only the closest one. To determine SSH at the time of SLA

measurement, we perform linear interpolation between the two nearest SSH observations. To remove bias and focus on the dynamic components, we subtract the mean values of SSH and SLA at each location. Although the measurements come from completely different sources, the visualizations indicate a strong correlation between SLA and SSH, with an average absolute difference of 4.0 cm.

    Despite the availability of SLA data from multiple satellites, its spatial and temporal coverage remains highly uneven.

Due to the repetitive nature of satellite orbits, SLA observations are concentrated along specific ground tracks rather than being uniformly distributed. Figure 3 visualizes the spatial distribution of SLA measurements, showing that the majority of observations originate from a limited number of tracks, leaving vast regions with very few SLA measurements.

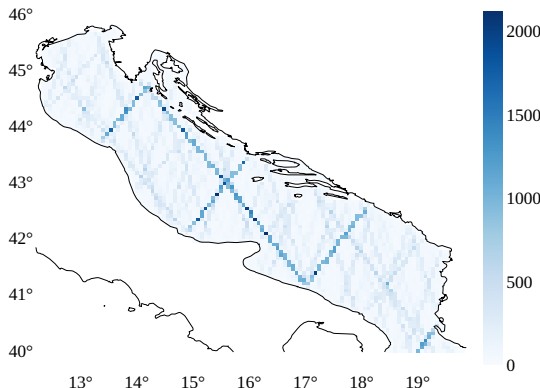

**Figure 3.** Spatial distribution of count of SLA measurements, where values in the image represent the number of SLA measurements recorded between 2000 and 2020.

    Figure 4 illustrates the frequency of swaths between the years 2000 and 2020. Given the high variability of sea level dynamics, SLA data presents an extremely sparse source of ground truth. On average, 2.4 swaths per day are recorded, meaning that

in 90 % of hourly intervals, no SLA measurements are available beyond tide gauge measurements. In the remaining 10 % of cases, a single swath is present, covering an average distance of 156 km, corresponding to approximately 25 grid points. In our setup, the Adriatic basin covers of 3,413 grid points, meaning that a single swath typically covers less than 1 % of the total number of grid points.



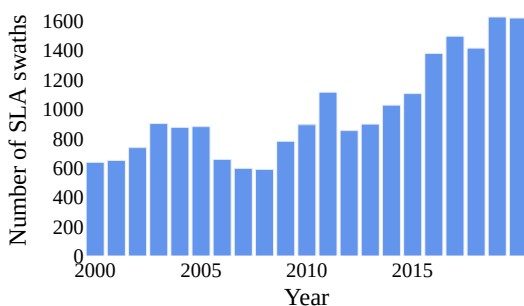

**Figure 4.** Number of SLA swaths per year from 2000 to 2020. On average, only 2.4 swaths were recorded daily, which highlights the sparsity of the SLA measurements.

## 2.2 HIDRA-D model architecture

The primary objective of the HIDRA-D model is to forecast dense sea levels across the basin. A key challenge here is that available inputs are limited to past SSH measurements, which are sparsely located at tide gauge positions, and can be supplemented with geophysical variables such as wind, air pressure, sea surface temperature (SST), and waves. To address the need for a comprehensive understanding of the sea basin's state from these sparse inputs, we leverage our previously developed multipoint SSH prediction model, HIDRA3 (Rus et al., 2025d), described in Sect. 2.2.1. HIDRA3 has proven to be effective

in generating a rich latent representation of the sea basin's state, which can subsequently be decoded into accurate SSH predictions at multiple tide gauge locations, including those that were absent from the input. In HIDRA-D, we take this latent representation derived from HIDRA3, along with the aforementioned geophysical features, as the foundation to forecast dense sea level values (see Fig. 5 for architecture). For this purpose, we introduce the Dense decoder module (detailed in Sect. 2.2.2). However, directly forecasting dense sea level fields presents a significant challenge: the output dimensionality is very large,

which would typically necessitate a model with a large number of parameters. Furthermore, observed sea level variations are not characterized by an equal distribution of frequencies; rather, low-frequency components account for the majority of sea level fluctuations. Consequently, to make the prediction task more tractable and efficient, we design the Dense decoder module to forecast only these dominant low-frequency components along with their corresponding phases. This forecasting is performed in the 2D Fourier domain for each of the 72 hourly prediction points. The final dense predictions are then efficiently

reconstructed by applying the inverse 2D discrete Fourier transform (2D IDFT) to these predicted Fourier coefficients.

### 2.2.1 HIDRA3 module

The first step in HIDRA-D's architecture is the production of SSH point forecasts at all tide gauge locations. For this, we use the HIDRA3 model (Rus et al., 2025d). HIDRA3 takes as input the past 72 h of tide gauge measurements, past and future 72 h of tidal predictions, and past and forecasted 72 h of geophysical variables. Based on these inputs, it generates 72 h SSH

forecasts at the tide gauge locations. We selected HIDRA3 due to its strong performance and its ability to handle missing past





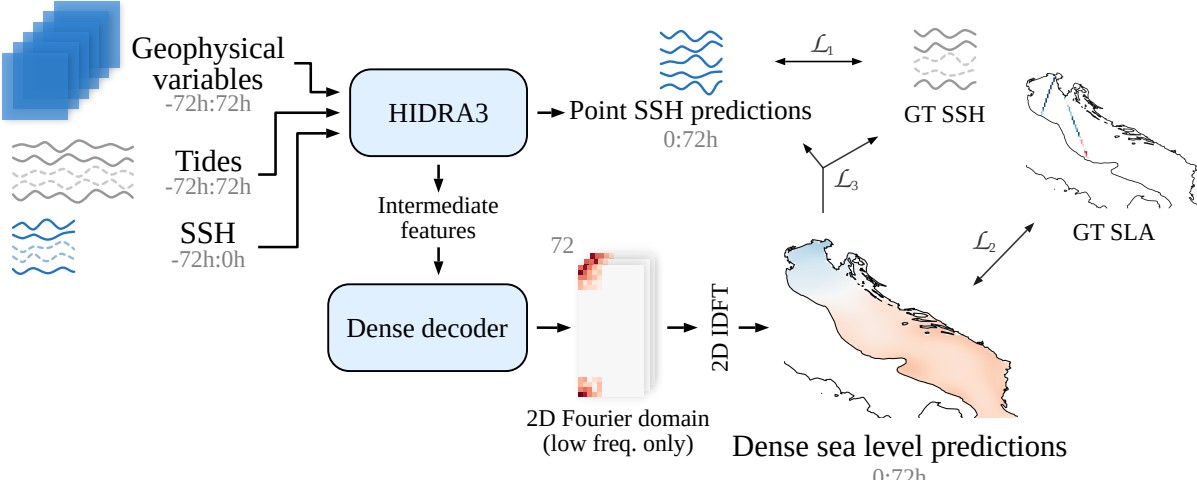

**Figure 5.** The HIDRA-D architecture. Dashed curves in SSH data indicate potential unavailable tide gauge data. The notation $a{:}b$ indicates hourly data points from the interval $(a, b]$, while the prediction point is at the index 0.

SSH measurements while consistently providing predictions for all locations. This is crucial for training, as tide gauge data availability varies (see Sect. 2.1).

While the HIDRA3 module yields predictions for specific tide gauge locations, obtaining dense predictions requires a different approach. Instead of directly using the SSH point predictions, we utilize the intermediate features generated by HIDRA3.
Specifically, we extract two key sets of features.

First, we utilize the geophysical feature vector. This vector is produced by HIDRA3's Geophysical Encoder module, which processes meteorological and oceanographic data fields (wind, air pressure, sea surface temperature, and waves) for the entire forecast region. The input data, covering a 144-hour window, is processed through a series of 3D and 1D convolutions to capture both spatial and temporal patterns, ultimately producing a single, comprehensive feature vector $s_{\text{geo}}$ of dimensionality
8192. This vector represents the overall geophysical state of the system.

Second, for each tide gauge $i$, we employ the per-tide gauge feature vector, denoted as $\hat{x}_i$. These vectors, each with a dimensionality of 1024, are generated within HIDRA3's SSH regression module. Their function is to provide location-specific context for the final forecast. The generation of $\hat{x}_i$ is robust to data availability. If recent SSH measurements for tide gauge $i$ are available, its specific feature vector is computed from that data. However, if measurements are missing, the vector is instead
approximated from a joint state vector that aggregates information from all other available tide gauges. This ensures that a unique and informative feature vector $\hat{x}_i$ is available for every tide gauge, which is a key capability we leverage in HIDRA-D.

### 2.2.2  Dense decoder module

The architecture of the Dense decoder module is shown in Fig. 6. The module processes geophysical features $s_{\text{geo}}$ of size 8192 and $N$ station-specific feature vectors $\hat{x}_i$ of size 1024 each (see Fig. 6). Here, $N$ represents the number of tide gauge stations.



These features are concatenated into a single vector of size $8192 + N \times 1024$, with a dimension of 19,456 in our configuration. To reduce dimensionality and computational load, a dense (fully connected) layer with 8192 output channels is applied.

The output is passed through three additional dense layers, each with 8192 output channels. These layers incorporate Scaled exponential linear unit (SELU, Klambauer et al. (2017)) activation, dropout, and residual connections, as depicted in Fig. 6. Residual connections help preserve input features by enabling the network to learn additive, nonlinearly transformed features.

SELU is chosen over Rectified linear unit (ReLU, Nair and Hinton (2010)) for its superior gradient propagation characteristics, while dropout is employed as a standard regularization technique to prevent overfitting.

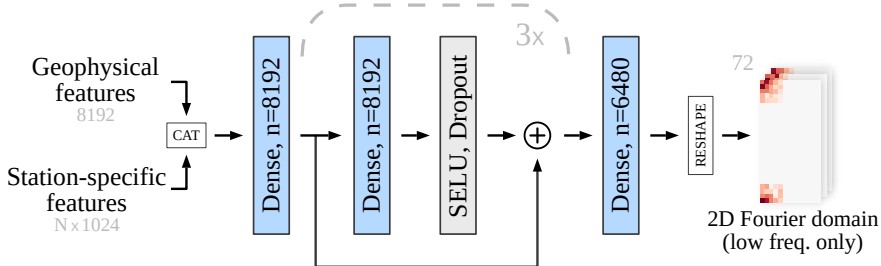

**Figure 6.** Structure of the Dense decoder module. Geophysical features and station-specific feature vectors are concatenated and processed through multiple dense layers that include SELU activation, dropout, and residual connections. The final output is reshaped, and the predicted values are assigned to the low-frequency components in the 2D discrete Fourier domain.

As discussed in Sect. 2.1.1, SLA data is extremely sparse and concentrated along specific ground tracks (Fig. 3). This sparsity can negatively impact model performance by potentially introducing unrealistically sharp sea level gradients at spatial scales considerably smaller than the local barotropic Rossby radius. Recognizing that large-scale forcings drive low-frequency

components that account for a significant portion of sea level variability, our modeling approach focuses exclusively on these components. The threshold for this frequency selection is determined by the barotropic Rossby radius. In the shallow northern Adriatic, with depths around 15 m, the Rossby radius is estimated to be approximately 120 km. Based on this, we set a spatial scale threshold of $\lambda_R = 150$ km, thereby modeling only processes with spatial scales larger than this value. The threshold is ablated in Sect. 3.6.

In the 2D discrete Fourier domain, the complex Fourier transform matrix $\mathbf{F}$ has dimensions $94 \times 115$. An element $\mathbf{F}_{ab}$ of this matrix corresponds to a pair of spatial frequencies $(k_x(a), k_y(b))$. These spatial frequencies (inverse spatial scales, $k = 2\pi/\lambda$) are computed as follows:

$$k_x(a) = \begin{cases} \frac{a}{W} k_x^s, & a = 0, 1, \ldots, \lceil \frac{W}{2} \rceil - 1 \\ \frac{a-W}{W} k_x^s, & a = \lceil \frac{W}{2} \rceil, \ldots, W - 1 \end{cases} \tag{1}$$

$$k_y(b) = \begin{cases} \frac{b}{H} k_y^s, & b = 0, 1, \ldots, \lceil \frac{H}{2} \rceil - 1 \\ \frac{b-H}{H} k_y^s, & b = \lceil \frac{H}{2} \rceil, \ldots, H - 1 \end{cases} \tag{2}$$





where $W = 115$ and $H = 94$ are the dimensions (number of grid points) of the spatial grid in the x and y directions, respectively. The terms $k_x^s \approx 2\pi/\Delta x$ and $k_y^s \approx 2\pi/\Delta y$ are derived from the grid spacings $\Delta x$ and $\Delta y$. For our domain, these values are $k_x^s \approx 1.16$ km$^{-1}$ and $k_y^s \approx 0.91$ km$^{-1}$.

To ensure that only wavelengths $\lambda > \lambda_R$ are represented in the output, the model predicts only those Fourier coefficients $\mathbf{F}_{ab}$
for which the corresponding spatial frequencies satisfy $|k_x(a)| < 2\pi/\lambda_R$ and $|k_y(b)| < 2\pi/\lambda_R$. Since the model predicts real-valued sea level fields, the Fourier matrix $\mathbf{F}$ must be Hermitian. Consequently, it is only necessary to predict approximately half of the Fourier coefficients, as the other half can be computed by transposition and conjugation. The set of Fourier coefficients that the Dense decoder module is tasked to predict, consists of elements forming $5 \times 5$ and $4 \times 5$ submatrices located in the corners of the full Fourier matrix $\mathbf{F}$ (see Fig. 6). The final dense layer of the Dense decoder module therefore outputs a vector
of 6480 features. This vector contains predictions for 72 forecast lead times, and for each lead time, 90 features are used to populate the real and imaginary parts of the elements within these $5 \times 5$ and $4 \times 5$ submatrices of $\mathbf{F}$. For each of the 72 temporal slices, the matrix $\mathbf{F}$ is transformed into a spatial field using an inverse 2D discrete Fourier transform (2D IDFT). The resulting field is then adjusted using the land-sea mask for the Adriatic Sea. The final spatial predictions are obtained by concatenating the 72 processed temporal slices, resulting in a grid of size $72 \times H \times W$.

## 2.3 Aligning SLA and tide gauges

A significant challenge in combining satellite altimetry (SLA) and tide gauge data is the lack of calibration between the two sensor types. As defined in Sect. 2.1.1, SLA represents the level relative to a reference geoid. In contrast, tide gauges measure SSH relative to a local vertical datum, often tied to a local reference ellipsoid. These local reference points are distinct for each tide gauge and their vertical offset from the geoid is generally not readily available. Consequently, without knowing the
transformation between tide gauge SSH measurements and satellite SLA measurements, tide gauge data cannot be directly used for supervising the training of a model designed to predict SLA.

To address this discrepancy, we define an intercalibration transformation for each tide gauge $i$, to convert it into SLA. The following model is used to transform the tide gauge measurements $y_i^{\text{gauge}}$ into SLA ($y_i^{\text{SLA}}$):

$$y_i^{\text{SLA}} = y_i^{\text{gauge}} + b_i, \tag{3}$$

where $y_i^{\text{gauge}}$ is the raw measurement from tide gauge $i$, and $b_i$ represents the unknown vertical displacement for that specific tide gauge relative to the geoid. This displacement is generally changing in time but considering the moderate deviations over timescales considered in this study, we approximate it by a constant.

Within our model framework, these unknown bias terms, $b_i$, are treated as learnable parameters and are estimated concurrently with the main model parameters during the training process. As a pre-processing step, both the SLA dataset and each
individual tide gauge time series are independently centered by removing their respective means. Consequently, the training process optimizes the bias parameters $b_i$ to align the centered tide gauge observations with the SLA measurements in their vicinity, effectively connecting the two data sources.





## 2.4 The loss functions

HIDRA-D is trained end-to-end using a multi-component loss function designed to effectively leverage diverse data sources
and provide targeted supervision. While the primary goal is to predict dense sea level maps, relying solely on satellite SLA
measurements (denoted as $y^{\text{SLA}}$) can be insufficient for robustly training the entire network, especially its backbone compo-
nents. To address this, we incorporate tide gauge SSH measurements (denoted as $y^{\text{gauge}}$), which, despite their spatial sparsity
and missing values, provide valuable supervisory signals.

To foster robust learning of the HIDRA3 module's internal representations, which form the foundation for subsequent dense
predictions, a dedicated loss term, $L_1$, is applied. This loss directly supervises the HIDRA3 module by comparing its SSH
predictions ($y_{\text{p}}^{\text{gauge}}$) to local SSH measurements from tide gauges ($y^{\text{gauge}}$):

$$L_1 = \overline{(y_{\text{p}}^{\text{gauge}} - y^{\text{gauge}})^2}. \tag{4}$$

Here, the overline symbol $\overline{(\dots)}$ denotes the mean taken over all timesteps and all available tide gauge stations. This targeted
supervision provides an additional training signal specifically for the HIDRA3 backbone.

The dense 2D sea level predictions $y_{\text{p}}^{\text{grid}}$ are primarily trained using satellite SLA observations. The loss term $L_2$ measures
the mean squared error between the dense SLA predictions ($y_{\text{p}}^{\text{grid}}$) and satellite SLA observations ($y^{\text{SLA}}$):

$$L_2 = \overline{(y_{\text{p}}^{\text{grid}} - y^{\text{SLA}})^2}. \tag{5}$$

As the dense predictions ($y_{\text{p}}^{\text{grid}}$) have an hourly resolution, we linearly interpolate between two neighboring predictions to
match the precise timestamps of the asynchronous SLA measurements from satellites. $L_2$ extends the learning beyond tide
gauge locations, leveraging the broader spatial coverage of satellite data.

To further enhance the training of the Dense decoder and utilize all available data, tide gauge data are also incorporated into
its supervision through the loss term $L_3$. This is particularly beneficial for improving predictions at coastal locations where
tide gauges are situated. $L_3$ measures the mean difference between the dense sea level predictions at tide gauge locations ($y_{\text{p},i}^{\text{grid}}$)
and the corresponding tide gauge information. Since tide gauge observations ($y^{\text{gauge}}$) typically measure local SSH relative to
a local datum, they must be transformed to be comparable with the network's predictions. Recall from Sect. 2.3, that this
transformation is achieved by adding learned station-specific displacements $b_i$ to the tide gauge SSH observations $y_i^{\text{gauge}}$,
effectively converting them into local SLA equivalents.

A challenge with tide gauge data is the frequent unavailability of measurements. To ensure that $L_3$ is defined over a com-
prehensive set of points and to provide continuous guidance, missing tide gauge observations $y_i^{\text{gauge}}$ are replaced by the cor-
responding point SSH predictions ($y_{\text{p},i}^{\text{gauge}}$) from the HIDRA3 module. This is possible because the HIDRA3 module outputs
predictions for all tide gauges, even when input measurements for some tide gauges are missing. Consistent with the treat-
ment of $y_i^{\text{gauge}}$, these HIDRA3 predictions are also transformed into the space of SLA by adding the same station-specific





displacement $b_i$. Thus, $L_3$ is defined as:

$$L_3 = \begin{cases} \overline{(y_{\text{p},i}^{\text{grid}} - (y_i^{\text{gauge}} + b_i))^2} & \text{where observation } y_i^{\text{gauge}} \text{ exists,} \\ \overline{(y_{\text{p},i}^{\text{grid}} - (y_{\text{p},i}^{\text{gauge}} + b_i))^2} & \text{where observation } y_i^{\text{gauge}} \text{ does not exist.} \end{cases} \tag{6}$$

This approach ensures that the Dense decoder module is consistently supervised at all tide gauge locations, leveraging either direct (transformed) observations or transformed HIDRA3 predictions as targets.

The final composite loss function, used for gradient backpropagation during the end-to-end training of the HIDRA-D model, is a weighted sum of these individual loss terms:

$$L = \alpha L_1 + \beta L_2 + \gamma L_3, \tag{7}$$

where the weights are empirically set to $\alpha = 100$, $\beta = 1$, and $\gamma = 1$. The high value of $\alpha$ ensures that training of the HIDRA3 module prioritizes $L_1$. Since $L_1$ is computed solely from the outputs of the HIDRA3 module, this prioritization effectively guides the backbone's learning without directly influencing the training gradients of the Dense decoder module from this specific term.

## 2.5 Training details

We use the AdamW optimizer (Loshchilov and Hutter, 2017) with a learning rate of $10^{-5}$ and a weight decay of 0.001. A higher learning rate of $10^{-3}$ is used for training the displacements $b_i$. To progressively decrease the learning rate by a factor of 100, we utilize a cosine annealing schedule (Loshchilov and Hutter, 2016). Parameters are initialized using a standard Xavier initialization (Glorot and Bengio, 2010), and parameters of the HIDRA3 module are initialized as described in Rus et al. (2025d). During training, tide gauge failures are simulated by randomly deactivating a subset of tide gauges with a probability

of 0.5. The model is trained for 50 epochs with a batch size of 128 data samples. Prior to training, all input data is standardized by subtracting the mean and dividing by the standard deviation. The mean is computed independently for each tide gauge location, while a single standard deviation is determined across all locations. Each geophysical variable and SLA undergoes independent standardization. Training requires approximately 12 h on a system equipped with an NVIDIA A100 Tensor Core GPU.

## 3  Results

### 3.1  NEMO model description

The numerical ocean model used in this paper is the Copernicus Marine Environment Monitoring Service (CMEMS) product MEDSEA_ANALYSISFORECAST_PHY_006_013 (Clementi et al., 2021), based on the Nucleus for European Modelling of the Ocean (NEMO) v4.2 (Madec, 2016). The Mediterranean Sea Physical Analysis and Forecasting model (MEDSEA_ANA-

LYSISFORECAST_PHY_006_013) spans a regular grid with a 1/24° (approximately 4 km) horizontal resolution and 141





| Model | MAE (cm) | RMSE (cm) | Bias (cm) |
| --- | --- | --- | --- |
| NEMO | 4.85 | 6.17 | 0.00 |
| HIDRA-D | **3.49** | **4.82** | −0.17 |

**Table 1.** Comparison of MAE, RMSE and bias between HIDRA-D and NEMO, based on SLA data over Adriatic basin during the testing period from June 2019 to the end of 2020. Bold values highlight the best performance. The bias for NEMO is zero, as an offset correction was applied to its forecasts.

vertical z*-levels with partial cells to accurately represent the model topography. It employs a staggered Arakawa C-grid with land area masking. Tidal forcing is represented by eight tidal constituents (M2, S2, N2, K2, K1, O1, P1, Q1). The model is forced at its Atlantic lateral boundary by the Global analysis and forecast product (GLOBAL_ANALYSISFORECAST_PHY-_001_024) and by a combination of daily climatological fields from a Marmara Sea model and the global analysis product in

the Dardanelles Strait. Atmospheric surface forcing is provided by the ECMWF deterministic model. The model was initialized from the World Ocean Atlas (WOA) 2013 V2 winter climatology on January 1, 2015. In situ vertical profiles of temperature and salinity from ARGO, Glider, and XBT, as well as SLA data from multiple satellites (including Jason 2 & 3, Saral-Altika, Cryosat, Sentinel-3A/3B, Sentinel6A, and HY-2A/B) are assimilated via the OceanVar (3DVAR) scheme. The hydrodynamic part of the model is coupled to the wave model WaveWatch-III. We refer the reader to Clementi et al. (2021) for further details.

NEMO computes sea level as a local deviation from the geoid, in theory making it directly comparable to SLA measurements in our setup. However, when analyzing the mean difference between SLA observations and NEMO forecasts (extracted at the SLA measurement locations and linearly interpolated to the corresponding timestamps), we find that, on average, SLA observations are 34.61 cm higher than NEMO forecasts in the training set period. To correct this systematic bias, we apply an offset of 34.61 cm to the NEMO forecasts.

In the analysis of sea level forecasts produced by NEMO, we use the same region defined by HIDRA-D. The region is subsampled from a $141 \times 185$ grid to a $94 \times 115$ grid to match the resolution of HIDRA-D. This subsampled version is also used when comparing the model to SLA data, ensuring that the metrics are computed on the same set of SLA measurements in the same spatial locations.

### 3.2   Evaluation of dense sea level predictions over the Adriatic

To assess the accuracy of dense sea level predictions, we compare them against SLA values (i.e., satellite swath measurements) on the test set, spanning the period from June 2019 to the end of 2020. Table 1 presents the mean absolute error (MAE) and root mean squared error (RMSE) for HIDRA-D and NEMO (Madec, 2016). Results indicate that overall HIDRA-D outperforms NEMO, with both MAE and RMSE being significantly lower. Furthermore, HIDRA-D exhibits a very low bias of −0.17 cm. Since we performed a bias correction of NEMO to align it with SLA values (see Sect. 3.1), its bias is effectively zero.

To visually compare the forecasts, we present two examples of dense forecasts generated by HIDRA-D and NEMO, along with the difference between these forecasts. Figure 7 illustrates the forecasts under calm atmospheric conditions, while Fig. 8

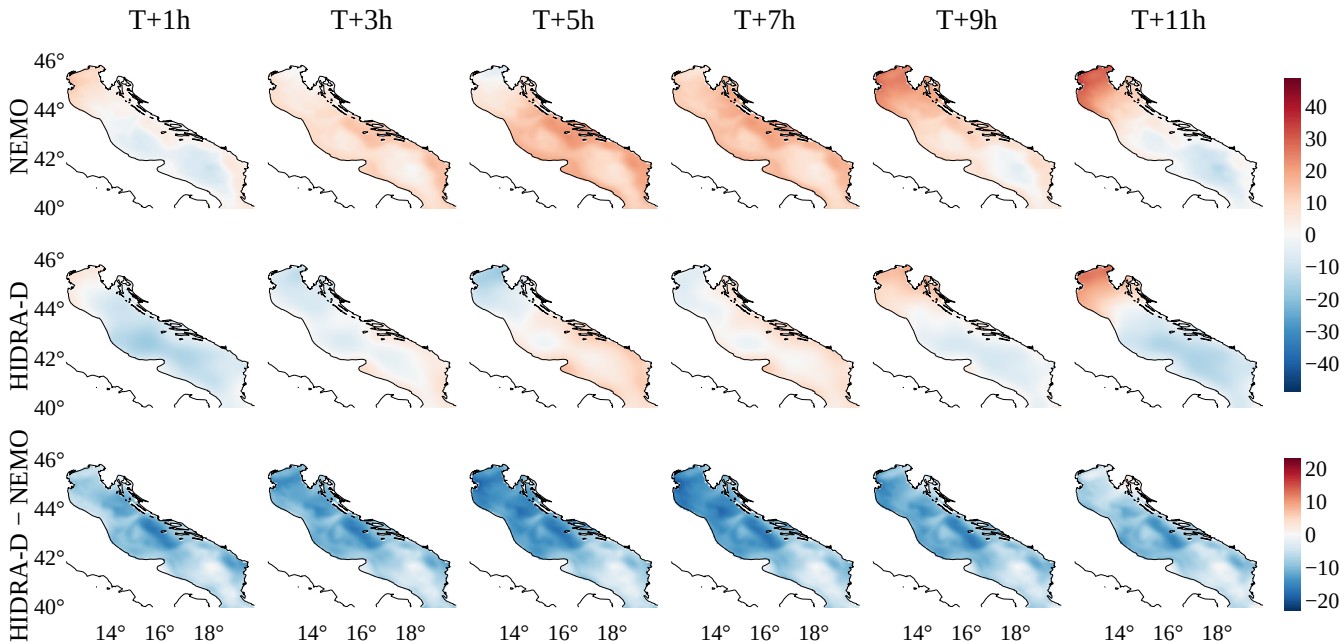

**Figure 7.** A subset of dense sea level predictions generated by HIDRA-D and NEMO under calm atmospheric conditions. The forecasts correspond to $T =$ November 4, 2020, 23:00, the units are cm. HIDRA-D produces a spatially smoother forecast compared to NEMO. The bottom row illustrates the difference between HIDRA-D and NEMO; note that it has a separate color bar.

depicts the forecasts of dense sea level during a storm surge event. Note that the HIDRA-D predictions are smoother, containing lower spatial frequencies than predictions from NEMO. This result is expected because HIDRA-D does not model processes with spatial scales below the barotropic Rossby radius, $\lambda_R$. These smaller-scale processes can be caused by other ocean

processes, such as ocean currents, which are not explicitly included in the HIDRA-D model. Despite this difference, both models generally produce forecasts of a comparable magnitude. Due to space limitations only selected parts of the forecasts are presented here, for visualizations of the entire forecast and additional forecasts, we direct the reader to the video supplement of the paper.

Figure 9 compares HIDRA-D and NEMO predictions with along-track SLA observations. Selected dates correspond to the

first day of each month from June 2019 to May 2020, with the longest swath in the basin chosen for each day. The forecasts were obtained for the same day as the SLA measurements, with model predictions interpolated to match the SLA measurement times. The results indicate that while both HIDRA-D and NEMO predictions are both smoother than SLA measurements, HIDRA-D predictions better capture the overall trend reflected in the measurements.



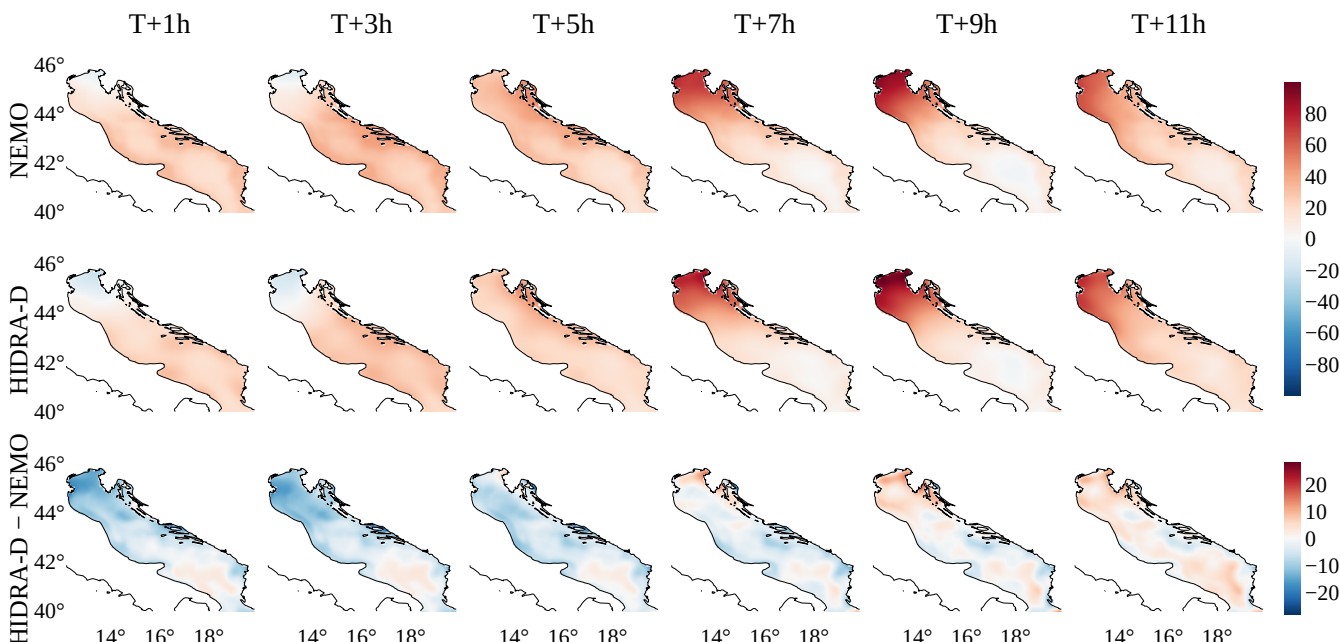

**Figure 8.** A subset of dense sea level predictions generated by HIDRA-D and NEMO during a storm surge event. The forecasts correspond to $T = $ October 14, 2020, 23:00, the units are cm. HIDRA-D produces a spatially smoother forecast compared to NEMO. The bottom row illustrates the difference between HIDRA-D and NEMO; note that it has a separate color bar.

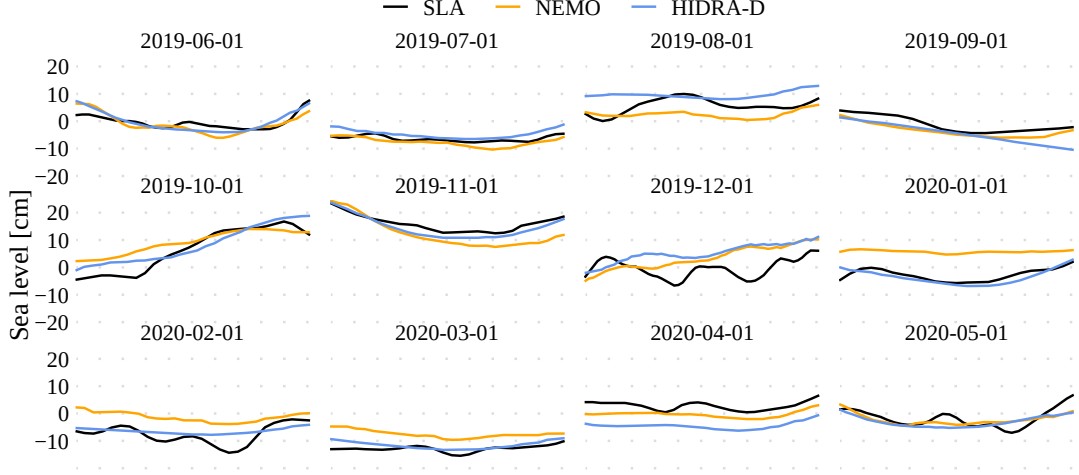

**Figure 9.** Comparison of along-track SLA with forecasts from HIDRA-D and NEMO. Predictions from both models were interpolated to match the SLA observation times.





### 3.3 Learned tide gauge displacements

The methodology described in Sect. 2.3 involves learning the vertical displacement, $b_i$, for each tide gauge $i$. A positive $b_i$ indicates that, on average, measurements from tide gauge $i$ are lower than SLA measurements in its vicinity. The resulting displacements for the different tide gauge locations are shown in Table 2. The values vary in both sign and magnitude, ranging from –0.8 cm at Koper to +5.6 cm at several Italian locations. This variation underscores the necessity of estimating these displacements individually for each tide gauge, as a single, uniform offset would not suffice to accurately align all tide gauge

records to the common SLA reference.

| Tide gauge location | Displacement ($b_i$) |
|---|---|
| Koper | –0.8 cm |
| Tremiti | 1.7 cm |
| Venice, Ravenna, Ancona | $\sim 5.6$ cm |
| Other tide gauges | $\sim 3.0$ cm |

**Table 2.** Learned vertical displacements ($b_i$) for each tide gauge. A positive $b_i$ indicates that, on average, measurements from tide gauge $i$ are lower than SLA measurements in its vicinity.

### 3.4 Performance along the coastal region

This section analyzes the HIDRA-D prediction accuracy at coastal locations to estimate the potential for its practical application in sea level prediction at locations without tide gauges. The evaluation setup involves removing, one at a time, one tide gauge from the total set of $N = 11$ tide gauges, and training a separate model for each excluded tide gauge. Each separate model for

each of the eleven tide gauges is therefore trained on 10 remaining tide gauges. When calculating scores or visualizing, we always use the model that was not trained on data from the corresponding tide gauge. When presenting results from such setup, we use the notation HIDRA-D$^N$, where $N = 11$ refers to the number of models trained and used in the evaluation. Note that since we do not have any information about tide gauges' local vertical datums, bias correction over the test set is applied for each excluded tide gauge and model separately.

The standard performance measures from Rus et al. (2023) are computed: mean absolute error (MAE), root mean squared error (RMSE), accuracy (ACC), bias, recall (Re), precision (Pr), and the F1 score. Table 3 shows the performance measures for all sea level values (*overall*) and separately for *low* and *high* sea level values (see Sect. 2.1 for definitions). The results indicate that, averaged over all tide gauges, HIDRA-D$^N$ achieves lower errors than NEMO, with a particularly significant reduction in error for low SSH values. In contrast, for high sea level values, NEMO outperforms HIDRA-D$^N$.

Figure 10 shows MAE scores for each tide gauge location for further insights. While HIDRA-D$^N$ achieves comparable error levels to NEMO at many locations, it exhibits significantly higher MAE at Koper, Venice, and Neretva for high sea level values. This discrepancy might occur for several reasons. Both Koper and Venice are situated in the northern Adriatic where storm surges are topographically amplified. Additionally, the availability of the SLA measurements in this area is extremely low (see





| | Model | MAE (cm) | RMSE (cm) | ACC (%) | Bias (cm) | Re (%) | Pr (%) | F1 (%) |
|---|---|---|---|---|---|---|---|---|
| Overall | NEMO | 3.82 | 4.95 | **95.25** | 0.00 | / | / | / |
| | HIDRA-D$^N$ | **3.61** | **4.83** | 95.24 | 0.00 | / | / | / |
| Low SSH | NEMO | 7.08 | 8.44 | 69.82 | 5.78 | 75.01 | 91.87 | 80.78 |
| Values | HIDRA-D$^N$ | **5.41** | **6.49** | **85.76** | **3.94** | **89.03** | **99.33** | **92.49** |
| High SSH | NEMO | **6.05** | **8.39** | **84.96** | **–4.77** | **93.86** | **98.76** | **96.08** |
| Values | HIDRA-D$^N$ | 8.61 | 11.26 | 69.82 | –7.69 | 88.67 | 98.44 | 93.10 |

**Table 3.** Performance comparison between HIDRA-D$^N$ and NEMO using tide gauge measurements across all SSH values, as well as separately for low and high SSH values. The reported scores represent averages over all tide gauge locations. The results indicate that HIDRA-D$^N$ achieves lower errors overall and particularly for low SSH values.

Fig. 3), resulting in a very limited number of training samples containing storm surges. In the case of Neretva, its location in a long, narrow bay further allows excitations of local seiches, which the current architecture of HIDRA-D is not designed to learn. These findings suggest that HIDRA-D$^N$ is better suited for forecasting sea level in open waters rather than in locations characterized by strong local effects, unless past training data from a tide gauge is available at the specific location.

It is important to note, however, that these conclusions apply to an arbitrary point on the coastline where no training data is provided. At training tide gauge locations, HIDRA-D performs marginally worse than HIDRA3. The average MAE across all stations increases from 2.42 cm to 2.70 cm. For high sea level values, the MAE increases from 4.06 cm to 4.99 cm. For low sea levels, the performance is similar, with an MAE of 3.30 cm for HIDRA3 and 3.37 cm for HIDRA-D.

Fig. 11 illustrates the forecasted SSH time series generated by HIDRA-D$^N$ and NEMO alongside tide gauge measurements under calm atmospheric conditions. Similarly, Fig. 12 presents these comparisons during a storm surge event. These figures provide a visual comparison of the model outputs against observations, and it can be observed that both models capture the sea level dynamics well. For additional visualizations, including further forecasts and failure cases, we refer the reader to the video supplement.

### 3.4.1 Dynamic local bias correction

This experiment evaluates the performance improvement in a specific scenario where a tide gauge station, previously unavailable, is newly installed. The introduction of this new tide gauge allows for the application of a dynamic local bias correction at test time, which is not possible before the station is installed. We evaluate the models using tide gauge observations by applying this dynamic local bias correction at each tide gauge rather than a constant one. We apply bias correction to the first 12 h of the forecast, following standard practice when using the NEMO model for tide gauge forecasting (Rus et al., 2023). To distinguish these results from those obtained in previous experiments, we denote the NEMO model with 12 h bias correction as NEMO$_b$. For comparability, we also apply a 12 h bias correction to HIDRA-D$^N$, referring to the adjusted model as HIDRA-D$_b^N$.





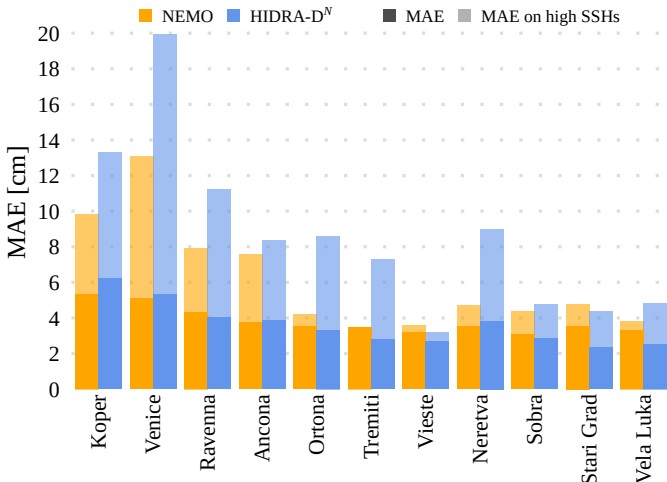

**Figure 10.** The overall MAE and the MAE for high SSH values on tide gauge measurements, computed for HIDRA-D$^N$ and NEMO. The models exhibit similar performance overall, while HIDRA-D$^N$ shows larger errors for high SSH values. Note that during training HIDRA-D$^N$ did *not* see any data from the respective station.

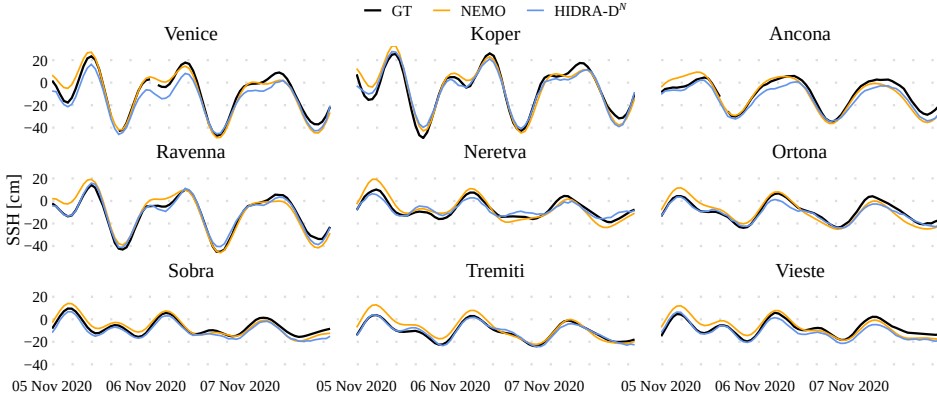

**Figure 11.** Prediction under calm atmospheric conditions. An example of a single forecast from November 4, 2020, at 23:00 for HIDRA-D$^N$ and NEMO, compared with tide gauge measurements. Note that during training HIDRA-D$^N$ did *not* see any data from the station on each panel.

380    Table 4 presents the average scores for both models across all tide gauge locations. The results indicate that NEMO$_b$ consistently exhibits lower errors than HIDRA-D$_b^N$. Figure 13 displays the MAE scores for all tide gauges, showing that HIDRA-D$_b^N$ again produces the highest errors in Koper, Venice, and Neretva, likely for the reasons discussed in Sect. 3.4. These findings suggest that when a newly installed tide gauge is available for bias correction, NEMO$_b$ provides more accurate predictions than HIDRA-D$_b^N$. However, as historical tide gauge data accumulates, training HIDRA-D on this location becomes feasible.



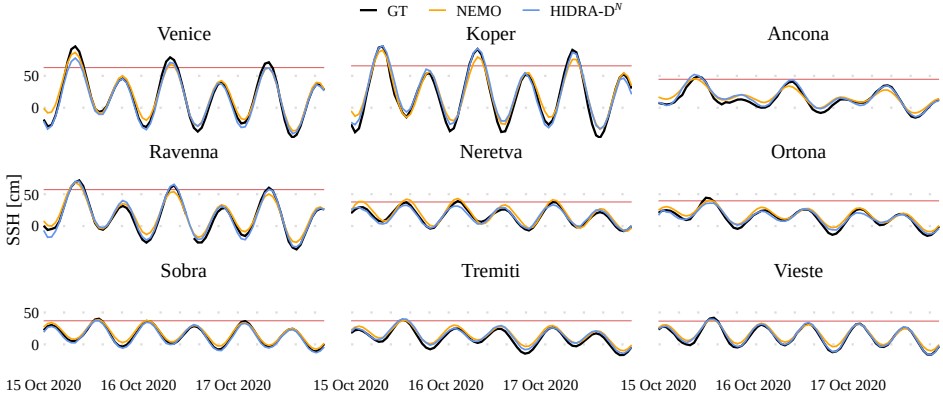

**Figure 12.** Prediction under storm surge atmospheric conditions. An example of a single forecast from October 14, 2020, 23:00 for HIDRA-D$^N$ and NEMO, compared with tide gauge measurements. The high SSH threshold is marked with a red line. Note that during training HIDRA-D$^N$ did *not* see any data from the station on each panel.

| | Model | MAE (cm) | RMSE (cm) | ACC (%) | Bias (cm) | Re (%) | Pr (%) | F1 (%) |
|---|---|---|---|---|---|---|---|---|
| Overall | NEMO$_b$ | **2.65** | **3.56** | **97.76** | –0.31 | / | / | / |
| | HIDRA-D$_b^N$ | 3.31 | 4.61 | 95.09 | **–0.04** | / | / | / |
| Low SSH | NEMO$_b$ | **4.19** | **5.23** | **92.91** | 2.88 | 94.04 | 99.92 | 96.39 |
| Values | HIDRA-D$_b^N$ | 4.92 | 6.31 | 87.74 | 3.78 | 91.25 | 98.80 | 94.15 |
| High SSH | NEMO$_b$ | **4.68** | **6.19** | **89.14** | –3.02 | 94.53 | 99.40 | 96.79 |
| Values | HIDRA-D$_b^N$ | 6.48 | 9.43 | 81.29 | –4.16 | 92.37 | 95.28 | 93.74 |

**Table 4.** Performance comparison between HIDRA-D$_b^N$ and NEMO$_b$, both bias-adjusted using the first 12 h of each forecast. The reported scores represent the average across all tide gauge locations. The results indicate that when data from a newly installed tide gauge is available, enabling dynamic bias correction, NEMO$_b$ achieves superior performance compared to HIDRA-D$_b^N$.

385    In fact, our recent work (Rus et al., 2025d) has shown that jointly trained on several tide gauges, HIDRA3 achieves excellent prediction accuracy at tide gauge locations even with a moderate amount of historical training data.

### 3.5   Removing regions of tide gauges

HIDRA-D is further evaluated by removing subsets of nearby tide gauges within the Adriatic basin. Specifically, we conduct two separate training experiments. In the first setup, we exclude tide gauge data from Koper, Venice, Ravenna, and Ancona, which are located in the northern Adriatic. We refer to this model as HIDRA-D$_S$, as it relies solely on measurements from tide gauges in the *southern* Adriatic. In the second setup, we use only the *northern* tide gauges, and denote the resulting model by HIDRA-D$_N$. Each model is then tested on the locations, that were excluded from its training.



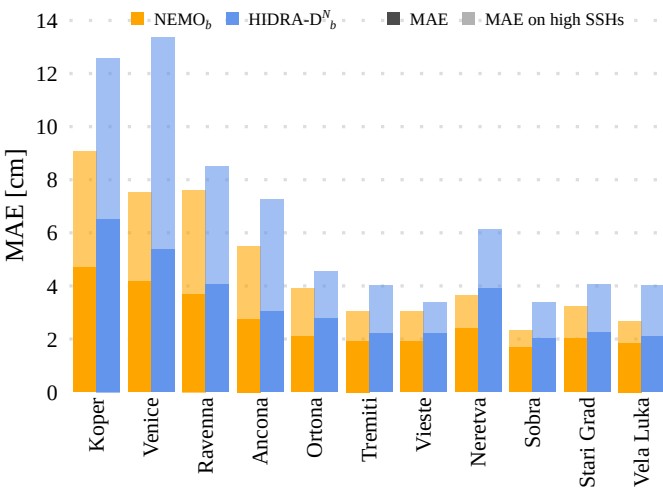

**Figure 13.** The overall MAE and the MAE for high SSH values in tide gauge measurements, computed for HIDRA-D$_b^N$ and NEMO$_b$. Both models were bias-adjusted using the first 12 h of each forecast. HIDRA-D$_b^N$ exhibits larger errors compared to NEMO$_b$.

Figure 14 presents the MAE scores, which are compared against those of HIDRA-D$^N$. The results indicate that the performance degradation remains minimal, despite the fact that the input SSH measurements originate from locations hundreds of kilometers away. This suggests that HIDRA-D effectively captures the overall dynamics of the Adriatic basin, even in scenarios with remote and spatially-limited tide gauge coverage.

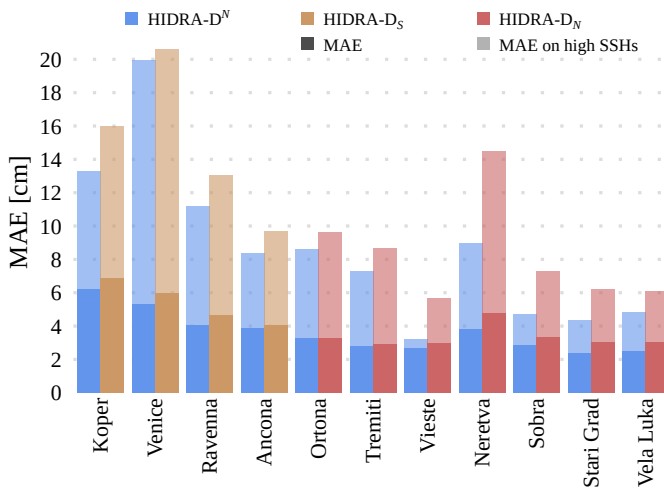

**Figure 14.** MAE increase when having only a subset of tide gauges as input. The model HIDRA-D$_S$ is trained using only tide gauges from the southern Adriatic, while HIDRA-D$_N$ is trained using only northern locations. The figure presents MAE scores for regions that were not included as input during training. The results show that, despite the exclusion of entire regions, the performance degradation remains minimal.





## 3.6  Influence of the spatial scale threshold

In Sect. 2.2.2, the northern Adriatic barotropic Rossby radius was used to define the spatial scale threshold as $\lambda_R = 150$ km. This threshold represents the lower limit for the forecasted wavelength in the dense output of HIDRA-D and determines the size of the non-zero element regions in the Fourier matrix $\mathbf{F}$, for $\lambda_R = 150$ equal to $5 \times 5$ and $4 \times 5$ submatrices. To investigate the influence of this hyperparameter, we conducted an ablation study by training model variants with one grid point smaller and larger submatrices. These variants are hereafter referred to as HIDRA-D$_{4\times4}$ and HIDRA-D$_{6\times6}$. For the cross-validation setup, they are denoted as HIDRA-D$^N_{4\times4}$ and HIDRA-D$^N_{6\times6}$.

The results of this ablation study are presented in Table 5. The evaluation on the SLA measurements shows that the MAE is largely unaffected by changes in the submatrix size. However, a different trend emerges in the cross-validation scenario, where models were evaluated on tide gauges that were excluded from training. In this case, our proposed model HIDRA-D$^N$, achieved the lowest MAE. This supports our selection of $\lambda_R$.

| Test data | Model | MAE (cm) |
|-----------|-------|----------|
| | HIDRA-D$_{4\times4}$ | 3.51 |
| Satellite SLA | HIDRA-D$_{6\times6}$ | 3.50 |
| | HIDRA-D | **3.49** |
| | HIDRA-D$^N_{4\times4}$ | 3.90 |
| Tide gauge SSH | HIDRA-D$^N_{6\times6}$ | 3.67 |
| | HIDRA-D$^N$ | **3.61** |

**Table 5.** Mean absolute error for different spatial scale thresholds, evaluated on SLA measurements and on excluded tide gauge measurements in a cross-validation setup. The best results are highlighted in bold.

## 4  Conclusions

This paper introduces HIDRA-D, a novel deep learning model for generating dense, two-dimensional sea level forecasts across an entire regional basin, representing a significant development over previous point-prediction models in the HIDRA family. HIDRA-D successfully integrates HIDRA3 module (Rus et al., 2025d) for point predictions at tide gauge locations with a new Dense decoder module that generates the low-frequency spatial components of the sea level field. Crucially, the model demonstrates a novel methodology for leveraging extremely sparse and unevenly distributed SLA data, combined with tide gauge observations, to achieve accurate two-dimensional basin-scale predictions. A key aspect of this integration is a new procedure for intercalibrating tide gauges and SLA, where the vertical displacement of each tide gauge is estimated as a learnable parameter during the model's training process, enabling the direct use of both satellite SLA and tide gauge SSH data for supervising.



Evaluated on the Adriatic Sea, HIDRA-D surpasses the NEMO general circulation model, achieving a 28.0 % reduction in MAE on SLA values. This demonstrates that deep learning is a viable, and often more accurate, alternative to computationally

expensive numerical models for sea level forecasting. Although HIDRA-D, like NEMO, captures large-scale sea level trends, it struggles to reproduce high-frequency local variations.

The model's performance is highest in open waters and degrades in coastal areas with complex bathymetry and limited data, such as Koper and Venice, especially during extreme sea level events. This limitation indicates that while HIDRA-D effectively learns basin-scale dynamics, resolving fine-scale, topographically-driven processes requires further development. Nonetheless,

the model exhibits remarkable robustness to a sparse tide gauge network, successfully capturing large-scale dynamics even when trained on remote stations. This feature is critical for applications in data-sparse regions.

Future research will focus on two main avenues. First, we aim to enhance the model's predictive skill by exploring methods to resolve higher-frequency spatial variations and better capture dynamics in complex coastal regions. Second, we will assess the generalizability of HIDRA-D by adapting and evaluating it in other ocean basins with diverse characteristics, such as larger

areas, different tidal regimes, and varying data availability.

*Code and data availability.* Implementation of HIDRA-D and the code to train and evaluate the model is available in the Git repository https://github.com/rusmarko/HIDRA-D (last access: 3 July 2025). The persistent version of the HIDRA-D source code is available at https://doi.org/10.5281/zenodo.15799686 (Rus et al., 2025b). HIDRA-D pretrained weights, predictions, geophysical training and evaluation data and SSH observations from Koper (Slovenia) are available at https://doi.org/10.5281/zenodo.15790578 (Rus et al., 2025c). Sea

level observations from Italian tide gauges are provided by The National Institute for the Environment Protection and Research (ISPRA) and are publicly available at the following address: https://www.mareografico.it (last access: 3 July 2025). Sea level observations from Neretva station are property of Croatian Meteorological and Hydrological Service (DHMZ) and are available upon request at the following address: https://meteo.hr/proizvodi_e.php?section=proizvodi_usluge¶m=services (last access: 3 July 2025). Sea level observations from Sobra, Vela Luka and Stari Grad (Croatia) are provided by the Institute of Oceanography and Fisheries (IOR) and are publicly available at the In-

tergovernmental Oceanographic Commission Sea Level Station Monitoring Facility (IOC SLSMF; http://www.ioc-sealevelmonitoring.org, last access: 3 July 2025). SLA data is publicly available through Copernicus Marine Service (CMEMS).

*Video supplement.* The video supplement presents animations of the complete dense sea-level forecasts from HIDRA-D and NEMO for both calm and storm surge conditions and includes additional forecast examples. The supplement also features animations of the forecasted SSH time series plotted against tide gauge measurements, corresponding to the events discussed in the main text. The video is available at

https://doi.org/10.5446/70892 (Rus et al., 2025a).

*Author contributions.* MR led the design of HIDRA-D. MK led the machine-learning research and contributed to the HIDRA-D design. ML led the oceanographic research, providing expertise on sea level geophysics. All authors collaborated on the manuscript.



*Competing interests.* The authors declare that they have no conflict of interest.

*Acknowledgements.* This research was made possible by the computational resources provided by the Academic and Research Network of
Slovenia (ARNES) and the Slovenian National Supercomputing Network (SLING) consortium through access to the ARNES computing
cluster. The authors also acknowledge the crucial role of the technical staff of the Italian National Institute for Environmental Protection and
Research (ISPRA), the Croatian Meteorological and Hydrological Service (DHMZ), the Institute of Oceanography and Fisheries (IOR), and
the Slovenian Environment Agency (ARSO) in maintaining the operational tide gauge network.

*Financial support.* Matjaž Ličer acknowledges the financial support from the Slovenian Research and Innovation Agency ARIS (contract
no. P1-0237). This research was supported in part by ARIS programme P2-0214 and project J2-2506.



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
