# Peer review of "HIDRA-D: deep-learning model for dense sea level forecasting using sparse altimetry and tide gauge data"

_EGUsphere, 2025_

## Author Comment (AC2)

Dear Editor and Reviewer,

Thank you for the constructive feedback regarding our submission. We have revised the manuscript based on the suggestions. Detailed point-by-point response to the reviewer's comments follows below.

**COMMENT 1:**

**The manuscript presents an interesting study on learning from observational datasets, and the authors propose a novel approach, building on their previous work on the HIDRA family, to generate dense grid predictions. The text is generally well written; however, the model architecture is not explained as clearly as expected, and my major concerns are regarding the evaluation, explainability, and generalization aspects. How can one transfer the model to other regions, and what are the limitations?**

**RESPONSE 1:**

We sincerely thank the reviewer for the positive feedback of our work and for raising many interesting points that helped us improve the paper. We address all the raised concerns carefully in the following. In particular, to summarize the general concerns stated in this comment:

*(1) The model architecture clarity concerns*: we address this in the revised paper by adding Problem definition subsection (see Response 2) and by further detailing the captions of the architecture figures (see Responses 9 and 10). For full transparency and reproducibility, the complete source code for training and evaluation is available.

*(2) The evaluation, explainability, and generalization concerns:* we agree that these are critical aspects of data-driven modeling. Unlike numerical models based on physical laws, HIDRA-D learns highly nonlinear statistical relations, which limits its interpretability. We have added a discussion on these trade-offs to the Introduction of the revised paper (see Response 3). Regarding generalization, we acknowledge that the current trained model is specialized for the Adriatic basin. However, the model architecture is general and designed to work for any location, though applying it to new regions requires retraining on local datasets. We have revised the Conclusions section to explicitly discuss these limitations (see below). Furthermore, we added a citation to a recent study where the HIDRA2 architecture was successfully applied to the Baltic basin, which gives us optimism regarding the transferability of HIDRA-D, its successor.

Updated paragraph in the Conclusions:

A limitation of data-driven approaches like HIDRA-D, compared to process-based numerical models, is their lack of physical interpretability. Future research will focus on addressing this and other open questions. First, we aim to enhance the model's predictive skill by exploring methods to resolve higher-frequency spatial variations and better capture dynamics in complex coastal regions. Second, we will assess the generalizability of HIDRA-D. While the architecture is general and designed to work for any location, application to new regions requires retraining on local datasets. We note that the HIDRA2 architecture was successfully applied in the Baltic basin along the Estonian coast (Barzandeh et al., 2025), performing well despite being originally developed for the Adriatic. This gives optimism that HIDRA-D would also perform well there, as it is its successor, although this remains to be thoroughly tested. We plan to adapt and evaluate the architecture in other ocean basins with diverse characteristics, such as larger areas, different tidal regimes, and varying data availability, to verify its transferability.

**COMMENT 2:**

**Suggesting sections 2.3 and 3.1 to a dataset section, along with the training dataset, to improve clarity and flow. Adding a problem definition subsection before the model architecture would be beneficial. The number of figures and tables is considerably higher than in a typical paper. I suggest reducing them in the main text and relocating additional information to an appendix to avoid distracting from the main flow of the research.**

**RESPONSE 2:**

As the reviewer suggested, we have reorganized the text by creating a unified *Data and experimental setup* section (merging Sect. 2.3 and Sect. 3.1) and adding a *Problem Definition* subsection prior to the model architecture (see below). We have also moved Figure 7 and Figure 11 to the Appendix to improve the readability of the main text.

**3.1 Formal problem definition of dense sea level prediction**

The goal of this study is to predict the temporal evolution of the ADT field, denoted as $\mathbf{Y}$, on a dense two-dimensional grid over the entire Adriatic basin. The target output is a sequence of hourly grid maps over a forecast horizon of $T = 72$ h. The spatial domain is defined by a grid of size $H \times W$ ($94 \times 115$), covering the latitudes 40.00°N to 45.87°N and longitudes 12.20°E to 18.85°E.

The model approximates the mapping function $\mathcal{F}$ such that $\mathbf{Y} = \mathcal{F}(\mathbf{X}_{\text{SSH}}, \mathbf{X}_{\text{geo}})$. The inputs consist of sparse SSH ($\mathbf{X}_{\text{SSH}}$), which encompasses the past 72 h of hourly SSH observations from $N = 11$ tide gauges along with their astronomical tide components, and geophysical forcing ($\mathbf{X}_{\text{geo}}$), a tensor containing the past 72 h and forecasted future 72 h of atmospheric and oceanic variables (e.g., wind, pressure, SST) derived from numerical models.

A fundamental challenge in this setting is that the ground truth for the dense output $\mathbf{Y}$ (satellite altimetry) is spatially sparse and temporally intermittent. Furthermore, the input tide gauge measurements ($y^{\text{gauge}}$) and the target satellite data ($y^{\text{ADT}}$) utilize different vertical datums. Therefore, the problem formulation includes the simultaneous estimation of a set of station-specific bias parameters, $\{b_i\}_{i=1}^{N}$, to align the inputs with the prediction target.

**COMMENT 3:**

**L36: This statement requires an appropriate reference. Also, it is appreciated to mention the limitations and challenges of data-driven models, such as stability and physical fidelity?**

**RESPONSE 3:**

As suggested, we have added the necessary references to support this statement, and we revised the text to discuss the limitations of data-driven models:

(Rus et al., 2025d, 2023; Žust et al., 2021)~. These models are designed to drastically reduce the computational cost of sea level forecasting while maintaining or even exceeding the accuracy of traditional numerical models, a trend increasingly observed in geophysical fluid dynamics domains (Bi et al., 2023; Lam et al., 2023).

However, data-driven approaches face their own challenges, particularly regarding training stability, interpretability, and the preservation of physical conservation laws, which often require careful hybrid modeling strategies (Irrgang et al., 2021). Nevertheless, active research within the machine learning communities on explainable artificial intelligence (XAI) (Samek and Müller, 2019, together with growing efforts to adapt these methods in geoscience (Mamalakis et al., 2022; Chen et al., 2025), offers a promising outlook. We expect that emerging findings from these subfields will increasingly contribute to the interpretability of deep learning methods in geophysics.

**COMMENT 4:**

**L41: There are several successful models for ocean emulation and forecasting. It is unclear why the authors primarily cite their own work to demonstrate the capabilities of data-driven models in ocean applications. Also, Rus et al. is cited repeatedly in some parts of the text, which is somewhat distracting.**

**RESPONSE 4:**

We thank the reviewer for these valid points. We have modified the introduction of the revised paper to contextualize our work within the broader field, we have added citations for major global initiatives (ORCA-DL, TianHai, XiHe) and relevant regional architectures (OceanNet, MedFormer, SeaCast) to properly expose the state-of-the-art in ocean emulation. We paste here the revised paragraph:

The evolution of HIDRA parallels a broader paradigm shift in ocean forecasting, where data-driven models are increasingly outperforming traditional numerical solvers. Recent global initiatives, such as ORCA-DL (Guo et al., 2025), TianHai (Niu et al., 2025), and XiHe (Wang et al., 2024), have successfully employed deep learning to capture 3D ocean dynamics and eddy-resolving features with high physical consistency. At regional scales, architectures like OceanNet (Chattopadhyay et al., 2024) have introduced physics-inspired neural operators for sea-surface height emulation, while MedFormer (Epicoco et al., 2025) and SeaCast (Holmberg et al., 2025) have demonstrated superior forecasting skills specifically within the Mediterranean Sea. Within this rapidly advancing context, our work has focused on the specific challenge of coastal sea level prediction. The initial HIDRA1 model (Žust et al., 2021) established that deep learning could predict sea surface height (SSH) at a single tide gauge with improved accuracy and vastly reduced computational costs compared to operational numerical model NEMO GCM (Ličer et al., 2020). Subsequent iterations, HIDRA2 (Rus et al., 2023) and HIDRA3 (Rus et al., 2025d), addressed early limitations by improving accuracy and utilizing data from neighboring operational stations to handle sensor failures. However, these models remained fundamentally limited to specific sensor locations, lacking the capability to predict sea levels in open waters, a gap that gridded approaches in the wider field have begun to address.

**COMMENT 5:**

**L57: The terminologies of sea level are mixed here. SLA and ADT are distinct terms. Some studies use these terms imprecisely; however, when SLA, ADT, MDT, and SSH are discussed together, their distinct definitions should be respected. SSH represents sea level relative to the reference ellipsoid. ADT corresponds to sea level relative to the geoid (i.e., SSH - geoidHeight) and is fundamentally different from SLA, which measures sea level relative to a given mean sea surface (i.e., SSH - MSS); and MDT = MSS - geoidHeight. Hence, it is recommended to use the appropriate terms consistently throughout the paper.**

**RESPONSE 5:**

We thank the reviewer for spotting this. We have revised the manuscript to ensure the terms SLA, ADT, and SSH are used correctly. For satellite data, we have replaced SLA with ADT throughout the manuscript when referring to the satellite observations, as the variable we use is the sum of SLA, MDT, and corrections.

We have retained the term SSH for tide gauge observations. This distinguishes the in-situ measurements, which are relative to local vertical datums (often tied to a local reference ellipsoid), from the satellite ADT, which is relative to the geoid.

**COMMENT 6:**

**L115: By adding MDT to SLA, we will have ADT? If instantaneous satellite data are required for this task, why are OTC and DAC applied?**

**RESPONSE 6:**

In standard CMEMS L3 altimetry products, the variable SLA_filtered is provided with ocean tides (OTC) and atmospheric forcing (DAC) *removed* (i.e., corrected). However, the objective of HIDRA-D is

to forecast the *total sea level* (i.e., what a tide gauge measures) to support coastal flood risk management. Therefore, we add the provided ocean_tide, MDT and DAC components back onto the SLA_filtered. Such handling of the altimetry data is the same as in the Mediterranean Copernicus ocean model NEMO. A personal communication with the colleagues at CMCC, the CMEMS Mediterranean NEMO model provider, confirmed that HIDRA-D predictions are comparable and compatible with NEMO model results.

We have added additional explanation to the manuscript:

> To enable supervised training of the network on the entire basin, the along-track sea level anomalies (SLA) from altimeter satellites were acquired from the Copernicus marine service product `SEALEVEL_EUR_PHY_L3_MY_008_061`. The SLA  variables are provided relative to a 20-y mean (1993–2012) with a 1 Hz (~7 km) sampling resolution. This dataset incorporates data from all available altimeter missions, including Sentinel-6A, Jason-3, Sentinel-3A, Sentinel-3B, Saral/AltiKa, Cryosat-2, Jason-1, Jason-2, Topex/Poseidon, ERS-1, ERS-2, Envisat, Geosat Follow-On, HY-2A, HY-2B, and others. The  ADT values used in this study are computed as the sum of  the provided variables: SLA_filtered, ocean_tide, mean dynamic topography (MDT) and dynamic atmospheric correction (DAC). Note that ocean_tide (ocean tide correction, or OTC) and DAC are explicitly added back to the anomaly. This effectively reverses the standard altimetry corrections, restoring the high-frequency tidal and atmospheric surge signals that were filtered out, thereby reconstructing the instantaneous total water level observed by tide gauges. This is consistent with how  ADT is treated in the CMEMS NEMO model for the purposes of data assimilation (Ali Aydogdu, CMCC, personal communication) and thus enables comparisons to the NEMO model (Clementi et al., 2021).

**COMMENT 7:**

**L126: It's not clear what adjustments were applied, and is it only for visualization? Weren't they applied for model training and evaluation?**

**RESPONSE 7:**

For the plots, we subtracted the mean so that positive/negative values represent points above/below the average. The same adjustment is applied for model training and evaluation. As detailed in Section *Aligning ADT and Tide Gauges*, we independently center both satellite ADT and tide gauge time series by removing their respective means as a standard pre-processing step.

We have updated the text to explicitly state that this visualization adjustment is consistent with the pre-processing used for training:

> For comparisons with hourly measurements from our model or NEMO, the time of  a satellite ADT measurement was not rounded to the nearest hour, recognizing that sea level can change rapidly. Instead, hourly time series from our model or NEMO were linearly interpolated to the exact time of each  satellite ADT observation. The spatial locations of the  satellite ADT measurements were binned into a grid with size $94 \times 115$ equal to the spatial output of our model. In cases where multiple  satellite ADT measurements fall within the same grid cell, the average of those measurements is taken. For visualization purposes,  ADT values are adjusted to have the mean equal to zero. This adjustment is consistent with the model training and evaluation setup, where mean removal is applied as a pre-processing step (see Sect. 2.3).

**COMMENT 8:**

**L154-155: This sentence is not clear. Is this HIDRA models' challenge or a general challenge?**

**RESPONSE 8:**

We intended to express that the high dimensionality of the output is a general challenge in dense field forecasting by deep learning, rather than a specific limitation of the HIDRA architecture. We have revised the sentence:

values (see Fig. 5 for architecture). For this purpose, we introduce the Dense decoder module (detailed in Sect. 3.2.2). However,  directly forecasting dense sea level fields is the very large output dimensionality, which would typically necessitate a model with a large number of parameters.

**COMMENT 9:**

**Fig.5: GT was not defined. L(s) and their arrows are confusing, perhaps require an explanation in the caption. Is block HIDRA3 frozen, fine-tuned, or jointly trained during the training of model HIDRA-D?**

**RESPONSE 9:**

We have updated the caption to explicitly define GT, and described the specific role of each loss term indicated by the arrows. HIDRA3 block is jointly trained with the rest of the model, we have added the statement to the caption to make this clear:

[Figure]

**Figure 5.** The HIDRA-D architecture. The model is trained end-to-end using a composite loss function: $\mathcal{L}_1$ supervises HIDRA3 point predictions using GT (ground truth) SSH; $\mathcal{L}_2$ supervises dense predictions using available satellite GT ADT; and $\mathcal{L}_3$ ensures consistency between the dense output and tide gauge data (using GT SSH where available, or HIDRA3 point predictions otherwise). Dashed curves in SSH data indicate potential unavailable tide gauge data. The notation $a:b$ indicates hourly data points from the interval $(a, b]$, while the prediction point is at the index 0.

**COMMENT 10:**

**Fig.6: How can we intuitively explain the 2D Fourier domain? The reshape block and how the data are transformed from physical space to Fourier space are not clear.**

**RESPONSE 10:**

Note that the Dense Decoder module *does not transform data from physical space to Fourier space*. Instead, the neural network uses the learned latent features to *directly predict the coefficients in the Fourier domain* - the final dense layer of the network outputs values, which represent the amplitudes and phases (real and imaginary parts) of the sea level's low-frequency spatial components. The Reshape block in Fig. 6 reorganizes the flat output vector (dimension 6,480) into the 3D tensor shape required to populate the low-frequency corners of the Fourier grid. Then, as shown in the broader architecture (Fig. 5), these coefficients are subsequently transformed into physical space predictions using 2D IDFT. Since the 2D IDFT is differentiable, this allows us to place supervision on the spatial reconstruction at sparse locations, while the error gradient flows backward through the IDFT to supervise the network's ability to directly predict the complete dense output in the frequency domain.

To make this clearer to the reader, we updated the text:

To ensure that only wavelengths $\lambda > \lambda_R$ are represented in the output, the model predicts only those Fourier coefficients $\mathbf{F}_{ab}$ for which the corresponding spatial frequencies satisfy $|k_x(a)| < 2\pi/\lambda_R$ and $|k_y(b)| < 2\pi/\lambda_R$. Since the model predicts real-valued sea level fields, the Fourier matrix $\mathbf{F}$ must be Hermitian. Consequently, it is only necessary to predict approximately half of the Fourier coefficients, as the other half can be computed by transposition and conjugation. Specifically, the output from a Dense decoder populates complex coefficients in $\mathbf{F}$ at the lowest spatial frequencies, which in our setup correspond to $5 \times 5$ and $4 \times 5$ regions in the corners of the matrix $\mathbf{F}$ (see Fig. 6). The final dense layer thus outputs a vector of 6480 features (comprising 90 real and imaginary components for each of the 72 forecast lead times). The remaining elements of $\mathbf{F}$, corresponding to higher spatial frequencies, are set to zero. For each of the 72 temporal slices, the matrix $\mathbf{F}$ is transformed into a spatial field using an inverse 2D discrete Fourier transform (2D IDFT). The resulting field is then multiplied by a binary land-sea mask of the Adriatic. The final spatial predictions are obtained by concatenating the 72 processed temporal slices, resulting in a grid of size $72 \times H \times W$.

Updates in the caption of Figure 6:

[Figure]

**Figure 6.** Structure of the Dense decoder module. Geophysical features and station-specific feature vectors are concatenated and processed through multiple dense layers that include SELU activation, dropout, and residual connections. The final output vector is reorganized (reshaped) into a tensor format, and where the predicted values are assigned as coefficients to the low-frequency components in the 2D discrete Fourier domain (ready for the Inverse DFT shown in Fig. 5).

**COMMENT 11:**

**L231: If b is due to the difference between the vertical reference surface, it should be constant or change due to vertical land movements at the location of tide gauges. Isn't it?**

**RESPONSE 11:**

We agree that the parameter $b_i$ represents the offset between the satellite geoid reference and the local tide gauge datum, which we assume to be physically constant. Any temporal variation in this parameter would indeed be due to slow geological processes. Our original phrasing (*generally changing in time*) was intended to acknowledge these potential long-term trends, but we recognize that it misleadingly implied high-frequency variability. We have revised the text to clarify that this offset is effectively constant by definition, with vertical land movements being the only (negligible) source of variation over the studied period:

> To address this discrepancy, we  perform a bias correction for each tide gauge $i$, to convert it into ADT. The following model is used to transform the tide gauge measurements $y_i^{\text{gauge}}$ into  ADT $(y_i^{\text{ADT}})$:
>
> $$y^{\text{\sout{SLA}ADT}}_i = y_i^{\text{gauge}} + b_i, \tag{1}$$
>
> where $y_i^{\text{gauge}}$ is the raw measurement from tide gauge $i$, and $b_i$ represents the unknown vertical  bias for that specific tide gauge relative to the geoid. This displacement primarily represents the offset between the vertical datums, which we assume to be physically constant over the timescales of this study. While vertical land movements can induce slow temporal variations in this offset, these changes are not taken into account here; therefore, we approximate  $b_i$ as a constant.

**COMMENT 12:**

**Table 1: Could you also add the performance of Nemo and HIDRA-D against the tide gauge over the testing period?**

**RESPONSE 12:**

We thank the reviewer for this suggestion. While we agree that comparing performance against tide gauges is essential, we have chosen to keep the tide gauge evaluation separate from Table 1. The results in Table 1 utilize the full HIDRA-D model trained on all available tide gauges to evaluate dense basin-scale performance against satellite data. However, the experiment, where HIDRA-D is evaluated against tide-gauges, tests ability to make accurate predictions for new locations without tide-gauge data available for training – this requires a leave-one-out approach (denoted as HIDRA-D[N] in the manuscript) to ensure the network is not tested on the location which was available during training. Mixing results from the full network and the leave-one-out networks in a single table could be misleading, as they were trained on different datasets and under different setups.

To ensure the readers can immediately locate this information, we have added a sentence explicitly referencing the later section:

To assess the accuracy of dense sea level predictions, we compare them against  ADT values (i.e., satellite  along-track measurements) on the test set, spanning the period from June 2019 to the end of 2020. Table 1 presents the mean absolute error (MAE) and root mean squared error (RMSE) for HIDRA-D and NEMO (Madec, 2016) based on satellite ADT data. The performance of the model against in situ tide gauge measurements requires a specific cross-validation setup to avoid data leakage and is detailed separately in Sect. 4.3 and Table 3. Results indicate that overall HIDRA-D outperforms NEMO, with

| Model | MAE (cm) | RMSE (cm) | Bias (cm) |
|---|---|---|---|
| NEMO | 4.85 | 6.17 | 0.00 |
| HIDRA-D | **3.49** | **4.82** | −0.17 |

**Table 1.** Comparison of MAE, RMSE and bias between HIDRA-D and NEMO, based on  satellite ADT data over Adriatic basin during the testing period from June 2019 to the end of 2020. Bold values highlight the best performance. The bias for NEMO is zero, as an offset correction was applied to its forecasts. The performance of the model against in situ tide gauge measurements requires a specific leave-one-out setup to avoid data leakage and is detailed separately in Sect. 4.3 and Table 3.

**COMMENT 13:**

**Fig. 7 & 8: It would be helpful to indicate the RMSE value for each panel in the bottom row.**

**RESPONSE 13:**

As suggested, we have calculated the root mean square difference (RMSD) between the HIDRA-D and NEMO forecasts, these values are now explicitly indicated in each panel of the bottom row:

[Figure]

**Figure 8.** A subset of dense sea level predictions generated by HIDRA-D and NEMO during a storm surge event. The forecasts correspond to $T =$ October 14, 2020, 23:00, the units are cm. HIDRA-D produces a spatially smoother forecast compared to NEMO. The bottom row illustrates the difference between HIDRA-D and NEMO; note that it has a separate color bar. The root mean squared difference (RMSD) between the two models is indicated in each panel.

[Figure]

**Figure B1.** A subset of dense sea level predictions generated by HIDRA-D and NEMO under calm atmospheric conditions. The forecasts correspond to $T =$ November 4, 2020, 23:00, the units are cm. HIDRA-D produces a spatially smoother forecast compared to NEMO. The bottom row illustrates the difference between HIDRA-D and NEMO; note that it has a separate color bar. The root mean squared difference (RMSD) between the two models is indicated in each panel.

**COMMENT 14:**

**Sec. 3.2: I suggest adding (or replacing with Fig. 7 or 8) the RMSE contour of HIDRA-D against Nemo for the lead times over the test dataset.**

**RESPONSE 14:**

We have included a new figure (see below) showing the spatial distribution of the RMSD between HIDRA-D and NEMO for lead times T+1 h, T+24 h, T+48 h, and T+72 h. The results indicate that discrepancies are highest in the Northern Adriatic and remain stable across different forecast horizons.

To further analyze the spatial and temporal structure of the discrepancies between the models, Fig. 7 illustrates the root mean square difference (RMSD) between HIDRA-D and NEMO across the basin for different forecast lead times. The metric is computed over the entire test period. Visually, the highest RMSD values (reaching approx. 8 cm) are concentrated in the northern Adriatic, likely due to the complex shallow-water dynamics in that region. Comparisons with available satellite ADT measurements (latitude $> 43.5°$) confirm that while both models exhibit higher errors in this area, HIDRA-D performs better with an RMSE of 5.37 cm compared to 6.79 cm for NEMO. In contrast, the central and southern parts of the basin generally exhibit lower differences, mostly ranging between 4 and 6 cm. Notably, both the spatial pattern and the magnitude of the RMSD remain remarkably stable across all lead times, indicating that the divergence between the two models does not grow significantly as the forecast horizon extends from T+1 h to T+72 h.

[Figure]

**Figure 7.** Spatial distribution of the root mean square difference (RMSD) between HIDRA-D and NEMO forecasts for lead times T+1 h, T+24 h, T+48 h, and T+72 h, computed over the test period. Discrepancies are most pronounced in the shallow northern Adriatic and remain stable across all lead times.

**COMMENT 15:**

**L323: according to Fig. 7, visually, one can observe that the difference between NEMO and HIDRA-D contains processes greater than \lambda_R=150km. Suggest comparing the radially averaged power spectrum of NEMO and HIDRA-D to discuss the spatial scales that the model can capture.**

**RESPONSE 15:**

We agree that comparing the radially averaged power spectra of NEMO and HIDRA-D would be an excellent method to quantitatively verify which spatial scales both models capture and to confirm if they share similar energy distributions in the low-frequency domain.

We attempted to perform this spectral analysis as suggested; however, we encountered significant technical challenges due to the complex geometry of the Adriatic basin. The irregular land-sea mask introduces sharp edges that severely distort the Fourier spectrum when standard 2D FFT methods are applied. We attempted to mitigate this by inpainting the land areas (e.g., using a Gaussian kernel), but the resulting spectra were dominated by processing artifacts, showing energy at frequencies that are explicitly excluded from HIDRA-D by design. Since we could not produce a faithful spectral representation without these artifacts, we omitted the plot.

At first glance an alternative would be to compute 1D power spectrum along a given line of sight (along the long axis of the basin or across the basin) but here we encounter the problem that the basin is 800 km long and 170 km wide, again limiting the spatial scales in one of the two directions. Therefore we could unfortunately not find a way to accommodate the reviewer's remark in a satisfactory manner.

**COMMENT 16:**

**Table 3: For what lead time? As a question, is RMSE simply averaged over all tide gages, or RMSE=sqrt(mean(MSE_i)). The second form should be presented as RMSE total. I suggest adding the performance of a naïve baseline model for comparison with Nemo and HIDRA-D, in which the forecast for the next time step is simply the last observed value (i.e., yp_{t+1} = y_t).**

**RESPONSE 16:**

Table 3 presents the average performance computed over all forecast lead times, spanning from T+1 hour to T+72 hours. The reported RMSE value is the average of the individual RMSE values computed for each tide gauge station.

As the reviewer suggested, we have computed a naïve Persistence baseline ($y_{t+1} = y_t$). Results indicate that this baseline performs significantly worse than both NEMO and HIDRA-D (e.g., Overall MAE of 15.46 cm vs. 3.61 cm for HIDRA-D), which shows the complexity of the forecasting task and the skill of the dynamic models. Given this, we chose not to include it in the revised version of the manuscript but provide the values here for reviewer's reference.

| | Model | MAE (cm) | RMSE (cm) | ACC (%) | Bias (cm) | Re (%) | Pr (%) | F1 (%) |
|---|---|---|---|---|---|---|---|---|
| | Persistence baseline | 15.46 | 19.38 | 44.86 | −0.84 | / | / | / |
| Overall | NEMO | 3.82 | 4.95 | **95.25** | 0.00 | / | / | / |
| | HIDRA-D$^N$ | **3.61** | **4.83** | 95.24 | 0.00 | / | / | / |
| | Persistence baseline | 36.71 | 38.51 | 5.28 | 36.71 | 5.18 | / | / |
| Low SSH Values | NEMO | 7.08 | 8.44 | 69.82 | 5.78 | 75.01 | 91.87 | 80.78 |
| | HIDRA-D$^N$ | **5.41** | **6.49** | **85.76** | **3.94** | **89.03** | **99.33** | **92.49** |
| | Persistence baseline | 27.76 | 32.11 | 22.44 | −27.15 | 23.22 | 17.68 | 19.89 |
| High SSH Values | NEMO | **6.05** | **8.39** | **84.96** | **−4.77** | **93.86** | **98.76** | **96.08** |
| | HIDRA-D$^N$ | 8.61 | 11.26 | 69.82 | −7.69 | 88.67 | 98.44 | 93.10 |

**Table 3.** Performance comparison between the Persistence baseline, HIDRA-D$^N$, and NEMO using tide gauge measurements. The evaluation covers all SSH values ("Overall"), as well as separate metrics for low and high SSH values. The reported scores are averaged over all forecast lead times (T+1 to T+72 h) and over all tide gauge locations. The results indicate that HIDRA-D$^N$ achieves lower errors overall and particularly for low SSH values, while both dynamic models significantly outperform the persistence baseline.

**COMMENT 17:**

**Fig. 10,13,14: I suggest including or replacing the plot with RMSE as a better index for performance assessment. I realised that black and gray colors in the legend refer to dark and light colors, but it's not a good way to show. Please show, e.g., dark and light orange for Nemo (and similarly for HiDRA-D) in the legend or simply remove the black and gray and mention it in the caption.**

RESPONSE 17:

We have replaced MAE with RMSE in Figures 10, 13, and 14. We removed the confusing color references from the legends and explicitly described the solid (overall RMSE) and semi-transparent (high SSH) distinctions in the captions:

[Figure]

**Figure 10.**  RMSE performance against tide gauge measurements  for HIDRA-D$^N$ and NEMO. Solid regions represent the overall RMSE, while semi-transparent regions indicate the RMSE for high SSH values. The models exhibit similar performance overall, while HIDRA-D$^N$ shows larger errors for high SSH values. Note that during training HIDRA-D$^N$ did *not* see any data from the respective station.

[Figure]

**Figure 12.**  RMSE performance against tide gauge measurements  for the HIDRA-D$_b^N$ and NEMO$_b$.  models. Solid regions represent the overall RMSE, while semi-transparent regions indicate the RMSE for high SSH values. A bias adjustment was applied using the first 12 h of each forecast. HIDRA-D$_b^N$ exhibits larger errors compared to NEMO$_b$.

[Figure]

**Figure 13.**  RMSE increase when having only a subset of tide gauges as input. The model HIDRA-$D_S$ is trained using only tide gauges from the southern Adriatic, while HIDRA-$D_N$ is trained using only northern locations. The figure presents  RMSE scores for regions that were not included as input during training. Solid regions represent the overall RMSE, while semi-transparent regions indicate the RMSE for high SSH values. The results show that, despite the exclusion of entire regions, the performance degradation remains minimal.

We have also updated the manuscript text to compare performance using RMSE at tide gauge locations used in training:

It is important to note, however, that these conclusions apply to an arbitrary point on the coastline where no training data is provided. At training tide gauge locations, HIDRA-D performs marginally worse than HIDRA3. The average  RMSE across all stations increases from  3.28 cm to  3.61 cm. For high sea level values, the  RMSE increases from 5.61 cm to  6.72 cm. For low sea levels, the performance is more similar, with an  RMSE of 4.24 cm for HIDRA3 and  4.41 cm for HIDRA-D.

**COMMENT 18:**

**Fig. 11: Are the time series hourly? Has any smoothing been applied to the tide gauge data? Have you calculated the correlation between tide gauges? I can see the neighbor tide gauges have similar behavior, so we can expect that excluding one tide gauge is unlikely to have a significant impact on model training.**

**RESPONSE 18:**

The data is hourly; we have made sure this information is in the *Problem Definition* section (see Response 2). The original data (1 min or 10 min resolution) was resampled to satisfy Nyquist constraints, i.e., the signals have been smoothed using a Gaussian kernel with σ = 25 min and then downsampled to an hourly resolution. See the updated section:

which results in a constant output value for an extended period of time, (ii) extreme outliers, and (iii) extreme jumps in the signals. Subsequently, the measurements were downsampled to hourly resolution. To prevent aliasing and reduce high-frequency noise, we applied a Gaussian smoothing kernel ($\sigma = 25$ min) using a weighted moving average prior to subsampling. The Gaussian weighted averaging was implemented with dynamic weight normalization to robustly handle missing time-steps in the raw high-frequency series. For each location, the astronomical tides in 1-year intervals were computed using the UTIDE Tidal Analysis package for Python (Codiga, 2011).

We agree that neighboring tide gauges in the Adriatic often exhibit similar behavior due to the basin's cohesive dynamics. To expose and validate this property, we have calculated the Pearson correlation coefficients between all tide gauge pairs and added a correlation matrix heatmap to the manuscript in the appendix:

**Appendix A: Tide gauge correlations**

To assess the redundancy of information provided by the tide gauge network, we computed the Pearson correlation coefficients between the SSH signals of all station pairs. Two distinct clusters with high internal correlation are visible (Fig. A1): the northern Adriatic group (Koper, Venice, Ravenna, Ancona) and the central/southern group.

[Figure]

**Figure A1.** Pearson correlation matrix of SSH measurements between different tide gauge locations. The stations are: Koper (KP), Venice (VE), Ravenna (RA), Ancona (AN), Ortona (OR), Tremiti (TR), Vieste (VI), Sobra (SO), Vela Luka (VL), Neretva (NE), and Stari Grad (SG). Two distinct clusters with high internal correlation are visible: the northern Adriatic group (KP, VE, RA, AN) and the central/southern group.

The plot confirms that there are high correlations between neighboring stations, particularly between northern (Koper, Venice, Ravenna, Ancona) and southern/central tide gauges. We agree that in the leave-one-out experiment, the model can indeed leverage information from highly correlated neighbors. However, this finding reinforces the importance of our *Removing regions of tide gauges* experiment. In that experiment, we deliberately removed correlated northern stations (Koper, Venice, Ravenna, Ancona). Despite this lack of correlated neighbors, HIDRA-D$_S$ still

performed with minimal degradation, proving that the model learns valid basin-scale dynamics and does not rely solely on local interpolation from nearby sensors.

Changes in *Training and testing datasets*:

The following tide gauges along the Adriatic coast are considered in this study for SSH measurements: Koper, Venice, Ancona, Ortona, Vieste, Neretva, Ravenna, Sobra, Stari Grad, Tremiti and Vela Luka (see Fig. 1). To characterize the inter-dependencies within this network, we analyzed the correlation between station records, observing strong clustering between neighboring stations (see Appendix A). Their SSH availability ranges from 15 % to 90 % during years 2000–2022 (Rus et al., 2025d), which has to be accounted for during training and testing, as the model is required to cast predictions also when data from

Changes in *Removing regions of tide gauges*:

**4.4 Removing regions of tide gauges**

HIDRA-D is further evaluated by removing subsets of nearby tide gauges within the Adriatic basin. Specifically, we conduct two separate training experiments.  As demonstrated in the correlation analysis (Fig. A1), the Adriatic tide gauges form two distinct clusters with high internal correlation: the northern group (Koper, Venice, Ravenna, Ancona) and the central/southern group. To rigorously test the model's ability to infer dynamics without relying on highly correlated neighbors, we first exclude tide gauge data from Koper, Venice, Ravenna, and Ancona. We

**COMMENT 19:**

**L402: The variants are not clear. Has the grid size changed after reshaping, or was the feature vector dimension modified and then reshaped to 4by4 instead of 5by5?**

**RESPONSE 19:**

The final spatial grid size (94 x 115) remains constant in all variants, the modification applies strictly to the cutoff frequency in the Fourier domain. Changing the size of the predicted Fourier submatrices requires modifying the output dimension of the final dense layer to match the changed number of Fourier coefficients. These coefficients are then placed into the corners of the full-size Fourier matrix (with high frequencies zeroed out) before applying the inverse transform. We have updated the Section:

**4.5 Influence of the spatial scale threshold**

In Sect. 3.2.2, the northern Adriatic barotropic Rossby radius was used to define the spatial scale threshold as $\lambda_R = 150$ km. This threshold represents the lower limit for the forecasted wavelength in the dense output of HIDRA-D and determines the size of the non-zero element regions in the Fourier matrix $\mathbf{F}$, which for $\lambda_R = 150$  km corresponds to $5 \times 5$ and $4 \times 5$ submatrices. To investigate the influence of this hyperparameter, we conducted an ablation study by training  two model variants where the dimensions of the predicted Fourier submatrices were increased or decreased by one element in each dimension. This modification required changing the output dimension of the final dense layer to match the number of coefficients in these resized submatrices. Note that the final spatial grid size $(94 \times 115)$ remains unchanged. These variants are hereafter referred to as HIDRA-D$_{4 \times 4}$ (utilizing $4 \times 4$ and $3 \times 4$ submatrices) and HIDRA-D$_{6 \times 6}$ (utilizing $6 \times 6$ and $5 \times 6$ submatrices). For the cross-validation setup, they are denoted as HIDRA-D$_{4 \times 4}^{N}$ and HIDRA-D$_{6 \times 6}^{N}$.

**COMMENT 20:**

**In the model definition, it would be worth mentioning that the tensor of Fourier components is padded to produce the desired output grid (?).**

**RESPONSE 20:**

The high-frequency components in the Fourier matrix F are set to zero to match the dimensions required for the inverse 2D discrete Fourier transform. We have added a sentence to clarify that:

 components for each of the 72 forecast lead times). The remaining elements of $\mathbf{F}$, corresponding to higher spatial frequencies, are set to zero. For each of the 72 temporal slices, the matrix $\mathbf{F}$ is transformed into a spatial field

**COMMENT 21:**

**L418: This is a strong claim, as the Nemo model is evaluated in an autoregressive (AR) mode, whereas HIDRA-D is assessed for single-step forecasting. I think these two evaluation settings are not directly comparable.**

**RESPONSE 21:**

We agree that the wording is too broad given the differences in methodology. However, we emphasize that both models are evaluated on exactly the same task: predicting the hourly evolution of sea level over a 72-hour horizon, given only the information available at T=0. Neither model has access to ground truth data during the forecast window. We have revised the text in both the Abstract and Conclusions to be strictly precise about the comparison. Instead of claiming general superiority, we explicitly mention the circumstances of comparison:

Changes in *Abstract*:

to learn the complex basin-scale dynamics of sea level. HIDRA-D achieves this by integrating a HIDRA3 module for point predictions at tide gauges with a novel Dense decoder module, which generates low-frequency spatial components of the sea level field in the Fourier domain, whose Fourier inverse is an hourly sea level forecast over a 3-day horizon.  When comparing 3-day forecasts against satellite absolute dynamic topography (ADT) data in the Adriatic, HIDRA-D  achieves a 28.0 % reduction in mean absolute error relative to the NEMO general circulation model. How-

Changes in *Conclusions*:

 When comparing 3-day forecasts of NEMO and HIDRA-D with satellite ADT measurements in the Adriatic Sea, HIDRA-D  achieves a 28.0 % reduction in MAE. This demonstrates that deep learning is a viable, and often more accurate, alternative to computationally expensive numerical

**COMMENT 22:**

**-- Some Minors:**

**Check the first line of the abstract. It's not aligned with the title regarding using satellite altimetry data.**

**RESPONSE 22:**

We have updated the abstract to explicitly mention that the model uses both sparse satellite altimetry and in situ tide gauge data:

This paper introduces HIDRA-D, a novel deep-learning model for basin scale dense (gridded) sea level prediction  using sparse satellite altimetry and in situ tide gauge data. Accurate sea level prediction is crucial for coastal risk management, marine operations, and sustainable development. While traditional numerical ocean models are computationally expensive, es-

**COMMENT 23:**

**L30: "when modeling using a numerical model", what about data-driven models? Is it only for numerical models?**

**RESPONSE 23:**

The inherent uncertainties in initial conditions and physical processes require a probabilistic approach in all types of models; we have revised the text to read *when modeling complex geophysical systems* to reflect this.

**COMMENT 24:**

**L31-33: This statement is somewhat overstated. Numerical models do not always rely on ensemble modeling, as this depends on the task. Also, ensembles vary not only in parameters but also in initial conditions, boundary conditions, and forcing fields (?).**

**RESPONSE 24:**

We agree. Numerical models are indeed often used deterministically, and ensemble spread is not generated just by varying parameters. Here is the revised text:

(Ferrarin et al., 2023; Bernier and Thompson, 2015; Mel and Lionello, 2014). Instead of a single, deterministic prediction, numerical  forecasting systems often employ ensemble modeling, generating multiple simulations with slightly varying initial conditions, forcings, or parameters to capture the envelope of possible sea level outcomes. While this approach

**COMMENT 25:**

**L38-41: It is expected that forecasting at a single point would be substantially faster than performing spatiotemporal Nemo forecasting?**

**RESPONSE 25:**

We agree that a significant speedup is naturally expected given the reduced scope of a single-point model compared to a numerical model covering the entire Mediterranean Sea. We have revised the text to claim *vastly reduced computational costs* instead of a specific multiplier, emphasizing the operational benefit rather than a direct computational comparison:

the specific challenge of coastal sea level prediction. The initial HIDRA1 model (Žust et al., 2021)  established that deep learning could predict sea surface height (SSH) at a single tide gauge  with improved accuracy and  vastly reduced computational costs compared to operational numerical model NEMO GCM (Ličer et al., 2020).

**COMMENT 26:**

**L61: "As a rule, "?**

**RESPONSE 26:**

We have removed the phrase.

**COMMENT 27:**

**L64-66: Please revisit these lines to ensure clarity.**

**RESPONSE 27:**

We have revised the lines:

 ADT measurements from the satellite altimeter are not calibrated with the SSH measurements at different tide gauges and, furthermore, the tide gauges are often not calibrated between each other, each reporting the sea level values relative to their local vertical datum. In this paper we propose a novel formulation that casts tide gauge and  ADT inter-calibration as part of the learning problem.  Specifically, the model estimates a vertical offset for each tide gauge, effectively aligning all stations to the common satellite-referenced ADT datum. This allows HIDRA-D to function in operational mode, where it generates dense, basin-scale ADT forecasts using only sparse tide gauge observations and atmospheric forcing.

**COMMENT 28:**

**The term "swath" is not commonly used for conventional satellite altimetry data, except for the SWOT dataset. Standard altimeters provide along-track measurements.**

**RESPONSE 28:**

We thank the reviewer for pointing this out. We have replaced all instances of *swath* with *track*, *ground track*, or *along-track*.

**COMMENT 29:**

**L222: "SLA represents the level relative to a reference geoid."?**

**RESPONSE 29:**

With the update from Response 5, the text now reads ADT instead of SLA.

---

## Author Comment (AC3)

Dear Editor and Reviewer,

Thank you for the constructive feedback regarding our submission. We have revised the manuscript based on the suggestions. Detailed point-by-point response to the reviewer's comments follows below.

**COMMENT 1:**

**Review of "HIDRA-D: deep-learning model for dense sea level forecasting using sparse altimetry and tide gauge data"**

**OVERVIEW**

**This paper presents an improvement to the existing HIDRA sea level forecasting model. The paper is interesting and rigorous, and the writing is clear. Aside from a few points and methodological steps that require further elaboration, the manuscript is well suited for publication. I suggest a minor revision.**

**RESPONSE 1:**

We sincerely thank the reviewer for the positive assessment of our work and the encouraging feedback. We have addressed the specific methodological questions and the suggestions regarding the discussion of high sea level performance.

**COMMENT 2:**

**HIGH SSH PERFORMANCE**

**From Table 3, it is apparent that HIDRA-DN performs markedly worse than NEMO for high SSH levels, which are usually the ones we care about most from an impact perspective. More discussion of this point is warranted than the existing note on line 354. This suggests that HIDRA-D is not "better" than NEMO at least for some applications, particularly coastal flood warning, a point which should be highlighted in the conclusions. Do you have an understanding or theory as to why the model performs worse on high SSH? Might there be a way to correct it that would not harm the model's overall skill?**

**RESPONSE 2:**

We agree that distinguishing between overall basin performance and the specific capability to predict coastal extremes is important, particularly for applications like flood warning. We have revised the manuscript to explicitly address this limitation in both the Discussion and Conclusions sections. Specifically, we expanded the discussion to detail why HIDRA-D$^N$ underperforms in these scenarios, listing three factors: the spatiotemporal sparsity of satellite ADT data which rarely captures transient storm surges; the regression-to-the-mean tendency of MSE-based training in data-sparse regimes; and the lack of explicit in-situ undisturbed water depths to resolve local topographic effects on sea level at untrained locations.

Regarding the possibility of correcting this behavior, we found that standard approaches like weighted loss functions (prioritizing extreme values) did not yield improvements in our experiments. We suspect this limitation is inherent to the current approach, specifically the reliance on sparse satellite altimetry which lacks sufficient dense ground truth for coastal extremes. We updated the conclusions to clarify that while HIDRA-D excels at basin-scale dynamics, traditional numerical models remain superior for coastal flood warnings at locations where no prior training tide-gauge data exists.

Changes in *Performance along the coastal region*:

In contrast, for high sea level values, NEMO outperforms HIDRA-D$^N$. Figure 10 shows RMSE scores for each tide gauge location for further insights. While HIDRA-D$^N$ achieves comparable error levels to NEMO at many locations, it exhibits significantly higher RMSE at Koper, Venice, and Neretva for high sea level values. This discrepancy likely arises from three connected reasons. First, the primary supervision for the dense field comes from satellite ADT data, which is spatiotemporally sparse (Fig. 3); the probability of a satellite track capturing the peak of a transient short-duration storm surge is low, leading to a training set imbalanced against extreme dense events. Second, deep learning models trained with mean squared error objectives tend to produce smoothed outputs to minimize global error, occasionally underestimating sharp peaks (regressing to the mean). Third, extreme values at specific locations are often driven by unresolved local topographic effects in bays and harbors. In the leave-one-out HIDRA-D$^N$ setup, the model must infer these local dynamics without ever seeing training data from that specific coastline geometry, whereas NEMO explicitly solves physical equations using high-resolution bathymetric grids. These findings suggest that HIDRA-D$^N$ is highly effective for open waters and general basin dynamics but faces limitations in resolving localized coastal extremes at locations where it has not been explicitly trained.

Changes in *Conclusions*:

computationally expensive numerical models for sea level forecasting. Furthermore, the model exhibits remarkable robustness to a sparse tide gauge network, successfully capturing large-scale dynamics even when trained on remote stations, which is critical for applications in data-sparse regions. However, although HIDRA-D, like NEMO, captures large-scale sea level trends, it struggles to reproduce high-frequency local variations and extreme peaks at untrained coastal locations. The model's performance is highest in open waters but degrades in coastal areas with complex bathymetry, such as Koper and Venice, specifically during extreme sea level events. Consequently, while HIDRA-D offers a computationally efficient alternative for basin-scale forecasting, it currently lags behind traditional numerical models like NEMO for specific applications such as coastal flood warnings at locations where no prior training data is available.

**COMMENT 3:**

**MINOR POINTS**

**Line 218: "The resulting field is then adjusted using the land-sea mask for the Adriatic Sea". Can you be more specific what this adjustment is?**

**RESPONSE 3:**

The adjustment is element-wise multiplication of the predicted 2D field with a static binary mask. Since the IDFT generates values for the entire rectangular grid, this masking step is necessary to set all values corresponding to land points to zero. Our specific mask is included in the published dataset.

The text now reads "The resulting field is then multiplied by a binary land-sea mask of the Adriatic."

**COMMENT 4:**

**Line 275: Describe in more detail how you arrived at these weights. Was there an objective hyperparameter tuning? Or hand-tuned? If so, what metric(s) were you trying to maximize during this tuning?**

**RESPONSE 4:**

Note that the weights were not determined through a numerical hyperparameter optimization process. Instead, they were chosen based on a specific architectural design choice to enforce a hierarchical prioritization of tasks. Our primary objective was to ensure that the HIDRA3 backbone, which generates the latent representations, retains its high accuracy for point-wise tide gauge predictions (supervised by $L_1$). If the weights were of similar magnitude, there would be a risk that the optimization process might degrade the accuracy of the HIDRA3 backbone (point predictions) in an attempt to minimize the dense reconstruction error ($L_2$ and $L_3$). By setting $\alpha = 100$, we impose a soft constraint that forces the backbone to prioritize the point-prediction task. This ensures that the dense predictions are constructed on top of a valid underlying state representation, rather than compromising that state to fit the sparse satellite data.

Regarding $\beta = 1$ and $\gamma = 1$: $L_2$ and $L_3$ generally operate on spatially distinct domains. $L_3$ provides supervision at fixed coastal points where satellite data is rarely available, while $L_2$ provides supervision in open water. Since they do not compete for the same spatial grid points in most iterations, equal weighting was sufficient.

We have revised the section in the manuscript to explain this design logic:

$$\mathcal{L} = \alpha\mathcal{L}_1 + \beta\mathcal{L}_2 + \gamma\mathcal{L}_3. \tag{7}$$

The weights were selected to enforce a hierarchical training structure: $\alpha = 100$, $\beta = 1$, and $\gamma = 1$. The significantly higher magnitude of $\alpha$ is a design choice intended to act as a soft constraint, ensuring that the HIDRA3 backbone prioritizes the accuracy of point-based SSH predictions ($\mathcal{L}_1$) above all else. This prevents the optimization of the dense reconstruction losses ($\mathcal{L}_2$ and $\mathcal{L}_3$) from degrading the quality of the underlying backbone representations. Consequently, the backbone learns primarily from the high-fidelity tide gauge data, while the Dense decoder adapts to these representations to satisfy the basin-wide constraints. The weights $\beta$ and $\gamma$ are set equally as $\mathcal{L}_2$ and $\mathcal{L}_3$ operate on largely spatially disjoint sets of points (sparse satellite tracks versus fixed tide gauge locations) and therefore do not require competitive weighting.

**COMMENT 5:**

**Line 281: "To progressively decrease the learning rate by a factor of 100, we utilize a cosine annealing schedule". Unclear. Does this mean that the starting and ending LR during training differ by a factor of 100? Or something else?**

**RESPONSE 5:**

As suggested, we clarified the statement in the revised paper. As the reviewer correctly inferred, the final learning rate is 1/100 of the initial learning rate. This decay factor applies proportionally to both parameter groups (the main model parameters starting at $10^{-5}$ and the bias parameters $b_i$ starting at $10^{-3}$), following standard annealing schedules (Loshchilov and Hutter, 2017). We have revised the text:

**3.4 Training details**

We use the AdamW optimizer (Loshchilov and Hutter, 2017) with  initial learning rate of $10^{-5}$ and a weight decay of 0.001. A higher initial learning rate of $10^{-3}$ is used for training the displacements $b_i$.  We utilize a cosine annealing schedule (Loshchilov and Hutter, 2016) to progressively decay the learning rate from its initial value $\eta$ to a minimum value of $\eta/100$ over the course of training. Parameters are initialized using a standard

**COMMENT 6:**

**Sec 2.5: More details required. Was a validation set used? Any early stopping conditions? How did you ensure overfitting did not occur?**

**RESPONSE 6:**

While a separate validation set was used during the development phase to tune hyperparameters, the final models presented in the paper were trained on the full training dataset to maximize the historical data availability. We did not employ dynamic early stopping for the final training, we relied on a fixed schedule of 50 epochs with a learning rate scheduler. To prevent overfitting, the model architecture and training process incorporate weight decay, dropout layers within the Dense

decoder, and a stochastic data augmentation strategy that randomly deactivates tide gauges, forcing the model to learn robust representations rather than memorizing specific sensor combinations.

We have expanded Section *Training details* to clarify our training protocol:

of 0.5. The model is trained for 50 epochs with a batch size of 128 data samples. Prior to training, all input data is standardized by subtracting the mean and dividing by the standard deviation. The mean is computed independently for each tide gauge location, while a single standard deviation is determined across all locations. Hyperparameters were tuned using a validation set separate from the test set. To maximize the data available for learning, the final models were trained on the full training dataset (see Sect. 2.1). Each geophysical variable and  satellite ADT data undergoes independent standardization. Training requires approximately 12 h on a system equipped with an NVIDIA A100 Tensor Core GPU.

**COMMENT 7:**

**Sec 3.1: This model description subsection seems better suited for the methods than the results.**

**RESPONSE 7:**

As suggested, we have reorganized the manuscript structure. The NEMO model description is now located in Section 2.4, within the Data and experimental setup section, separate from the section describing the model architecture.